# Permeation enhancer-induced membrane defects assist the oral absorption of peptide drugs

Kyle J. Colston[1], Kyle T. Faivre[2] & Severin T. Schneebeli [1,2] ✉

The passive membrane permeation of small-molecule drugs and small hydrophobic peptides is relatively well understood. In contrast, how long polar peptides can pass through a membrane has remained a mystery. This process can be achieved with permeation enhancers, contributing significantly to the oral transcellular absorption of important peptide drugs like semaglutide – the active pharmaceutical ingredient in Ozempic, which is used as Rybelsus in a successful oral formulation. Here we now provide a detailed, plausible molecular mechanism of how such a polar peptide can realistically pass through a membrane paired with the permeation enhancer salcaprozate sodium (SNAC). We provide both simulation results, obtained with scalable continuous constant $p$H molecular dynamics (C$p$HMD) simulations, and experimental evidence (NMR, DOSY, and DLS) to support this unique permeation mechanism. Our combined evidence points toward the formation of permeation-enhancer-filled, fluid membrane defects, in which the polar peptide can be submerged in a process analogous to quicksand.

Oral peptide drug formulations are highly desirable because they offer enhanced patient compliance compared to subcutaneous injections[1–4], while also preventing potential overdose issues that can arise with subcutaneous injections for GLP-1 agonists like semaglutide, which are projected to be used by over 9% of the American population by 2030[5]. However, the development of viable oral peptide formulations has been slow due to the low proteolytic stability and poor intrinsic gastric/intestinal permeability of most peptide drugs. For example, since the 1940s, the field has been striving to develop an oral formulation of insulin, but still, no widely adopted oral formulation has been discovered. In contrast, the passive membrane permeation of small molecules and macrocyclic peptides is reasonably well developed and understood, to the point where computational models are now able to predict the permeability of small molecule drugs[6,7] and macrocyclic peptides with predominantly hydrophobic side chains[8], especially when coupled with continuous constant $p$H molecular dynamics (C$p$HMD) models for small molecule drugs[6] (Fig. 1A) and sufficient conformational sampling for macrocyclic peptides[8] (Fig. 1B). At the same time

experimental evidence suggests that linear peptides like semaglutide (for the structure, see: Supplementary Figs. 5 and 6) can pass through epithelial cell layers with a transcellular absorption mechanism with the help of permeation enhancers like salcaprozate sodium (SNAC, see Fig. 2 for the structure)[9–12]. This approach has been adopted by Novo Nordisk to create one of the first oral formulations of a long, linear peptide drug (semaglutide), which is widely adopted on the market under the brand name Rybelsus[*5]. This formulation contains 7–14 mg of the peptide drug, co-formulated with ~400 mg of SNAC, representing a remarkable achievement in the oral peptide drug delivery space. Yet, the oral bioavailability of semaglutide with SNAC still remains low (<1%)[13], and rational design of improved permeation enhancer-assisted oral peptide formulations has been difficult due to the limited understanding of the corresponding absorption mechanism[14–17]. The current, partial knowledge about the stabilization/absorption mechanism of this formulation is mainly based on SNAC increasing the local $p$H around the Rybelsus® tablet[18], which, in turn, inactivates pepsin to protect semaglutide from proteolytic degradation in the

[1]Department of Industrial & Molecular Pharmaceutics, Purdue University, 575 Stadium Mall Drive, West Lafayette, IN 47907, USA. [2]James Tarpo Jr. and Margaret Tarpo Department of Chemistry, Purdue University, 575 Stadium Mall Drive, West Lafayette, IN 47907, USA. ✉e-mail: schneebeli@purdue.edu

**(A)** Ref. 6: Small-molecules

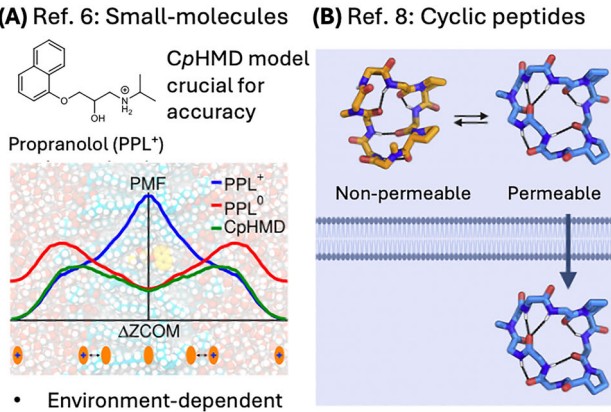

*Cp*HMD model crucial for accuracy

Propranolol (PPL⁺)

- Environment-dependent protonation states

**(B)** Ref. 8: Cyclic peptides

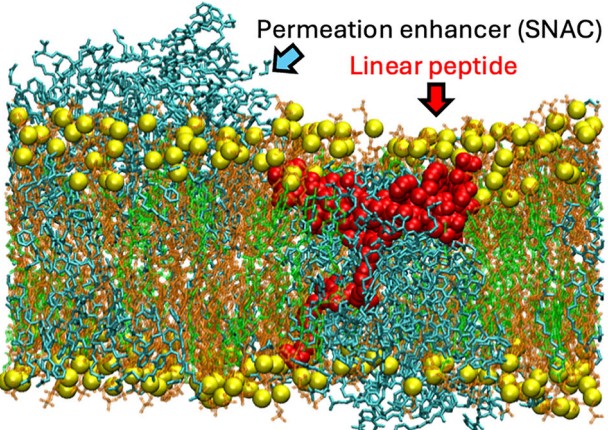

Non-permeable ⟷ Permeable

- Conformational Switching

**(C)** This Work: Polar linear peptide (semaglutide, 31 AAs)

Permeation enhancer (SNAC)

Linear peptide

Phosphate headgroup permeation barrier

**Fig. 1 | Different types of passive membrane permeation mechanisms.**
**A** Permeation of a small-molecule drug with environment-dependent protonation-state switching (ref. 6). **B** Permeation of a macrocyclic peptide drug with nonpolar side chains based on environment-dependent conformational switching (ref. 8). **C** Permeation-enhancer enabled incorporation of a linear, polar peptide drug (this work). Fig. 1A and B was reproduced in parts with permission from refs. 6,8.

stomach. This *p*H-induced proteolytic protection was experimentally determined to be effective at or above *p*H 5[19], which therefore represents a reasonable *p*H estimate for the local environment in the close vicinity of a Rybelsus® tablet (with local SNAC concentrations around a tablet as high as ~280 mM)[9]. Nevertheless, the detailed molecular mechanism by which SNAC enables semaglutide to permeate the gastric epithelial barrier has remained largely unknown. Most experimental evidence suggests[20] a transcellular absorption mechanism, with one of the current theories proposing that SNAC fluidizes the membrane, which then assists semaglutide in passing through the gastric epithelial cells[17]. Although initial computational studies have provided valuable insights into the interactions between SNAC and model membranes[21,22], no detailed molecular mechanism for semaglutide permeation has yet been verified. Specifically, it remains unclear how membrane fluidization can facilitate peptide permeation without damaging the epithelial cells.

In this work, we investigate the fundamental question of how long polar peptides can directly pass through a membrane with the help of permeation enhancers and describe a viable molecular mechanism for SNAC-enabled peptide permeation of semaglutide. By advancing toward this general goal, we aim to enhance the mechanistic understanding of oral peptide drug formulations (see Fig. 1C for an

example)[23]. Our results are obtained with accurate, all-atom *Cp*HMD simulations and are supported by experimental evidence obtained in CDCl₃ (as a well-established model for the hydrophobic membrane interior)[8,24–26] as well as with a CTAB micelle model (an alternative established membrane model in an aqueous environment)[27,28]. Notably, in our µs-long *Cp*HMD simulations, we observe semaglutide spontaneously incorporating into the membrane in the presence of the permeation enhancer SNAC, which explains how a long linear, and relatively polar peptide can get past the phospholipid barrier in a membrane in the presence of a permeation enhancer without damaging the integrity of the membrane. To account for all the weakly ionizable functional groups in our system (with ~400 ionizable groups in SNAC and 11 in semaglutide), we utilize a scalable *Cp*HMD model implemented in GROMACS[29]. This CpHMD model, which achieves ~80% of the performance speed of a regular MD simulation, is essential to provide accurate, environment-dependent *p*Kₐ values and environment-dependent charges for the complex peptide/permeation-enhancer/lipid-bilayer interactions in this system. Our simulations reveal a more even distribution of SNAC between the membrane and the water layer compared to simulations with a fixed protonation state model with SNAC always in the protonated state, enabling SNAC to act as a detergent in the water layer while at the same time facilitating peptide drug permeation within the membrane. Specifically, 1-µs unbiased *Cp*HMD simulations (likely among the largest *Cp*HMD models constructed to date) show that SNAC aggregates around semaglutide in both the interior and the exterior of the membrane. These aggregates form dynamic, partially fluid SNAC-filled membrane defects that can enable passive semaglutide permeation across the membrane by a mechanism reminiscent of quicksand. Additional support for dynamic SNAC aggregation in nonpolar environments is obtained through *Cp*HMD simulations in CH₂Cl₂, diffusion-ordered spectroscopy (DOSY) nuclear magnetic resonance (NMR), NMR titrations, dynamic light scattering (DLS) experiments in CDCl₃ (an established mimic of the hydrophobic membrane interior)[8,24–26], and ¹H-¹H NOESY NMR and DLS studies of SNAC interacting with CTAB micelles (as an alternative, well-established, soluble membrane model)[27,28]. Collectively, these results suggest a plausible molecular mechanism for passive membrane permeation with transcellular permeation enhancers, which we anticipate will aid in the rational design of future new oral peptide drug formulations.

## Results

The mechanism by which a long polar peptide like semaglutide integrates into a membrane while preserving the membrane's structural integrity amidst the influence of a permeation enhancer remains an enigmatic process. To investigate this pathway, we have now made use of all-atom *Cp*HMD simulations (with all the ionizable groups of the peptide and the permeation enhancers treated with a dynamic protonation state model). Notably, the *p*Kₐ values of ionizable sites can change drastically in or near a membrane[6] and/or during complexation/aggregate formation, and *Cp*HMD simulations are an accurate way to capture such environment-dependent effects on the charges of the peptides and the permeation enhancers. Foundational work by the Swanson lab has shown that such effects are key to accurately model the membrane permeation of small-molecule drugs[6]. Yet, prior *Cp*HMD simulations have been limited[29–34] to smaller systems with relatively few ionizable functional groups, mostly due to the unfavorable scaling of previous simulation approaches with the increasing number of ionizable sites. With the advent of scalable *Cp*HMD models[29], we have now been able to extend this type of simulation to larger systems. Our work likely represents one of the largest *Cp*HMD simulation systems run to date (with over 400 ionizable functional groups treated at the *Cp*HMD level). Below, the detailed results of these *Cp*HMD simulations are provided together with experimental evidence, all supporting our proposed quicksand-like permeation mechanism for polar peptides.

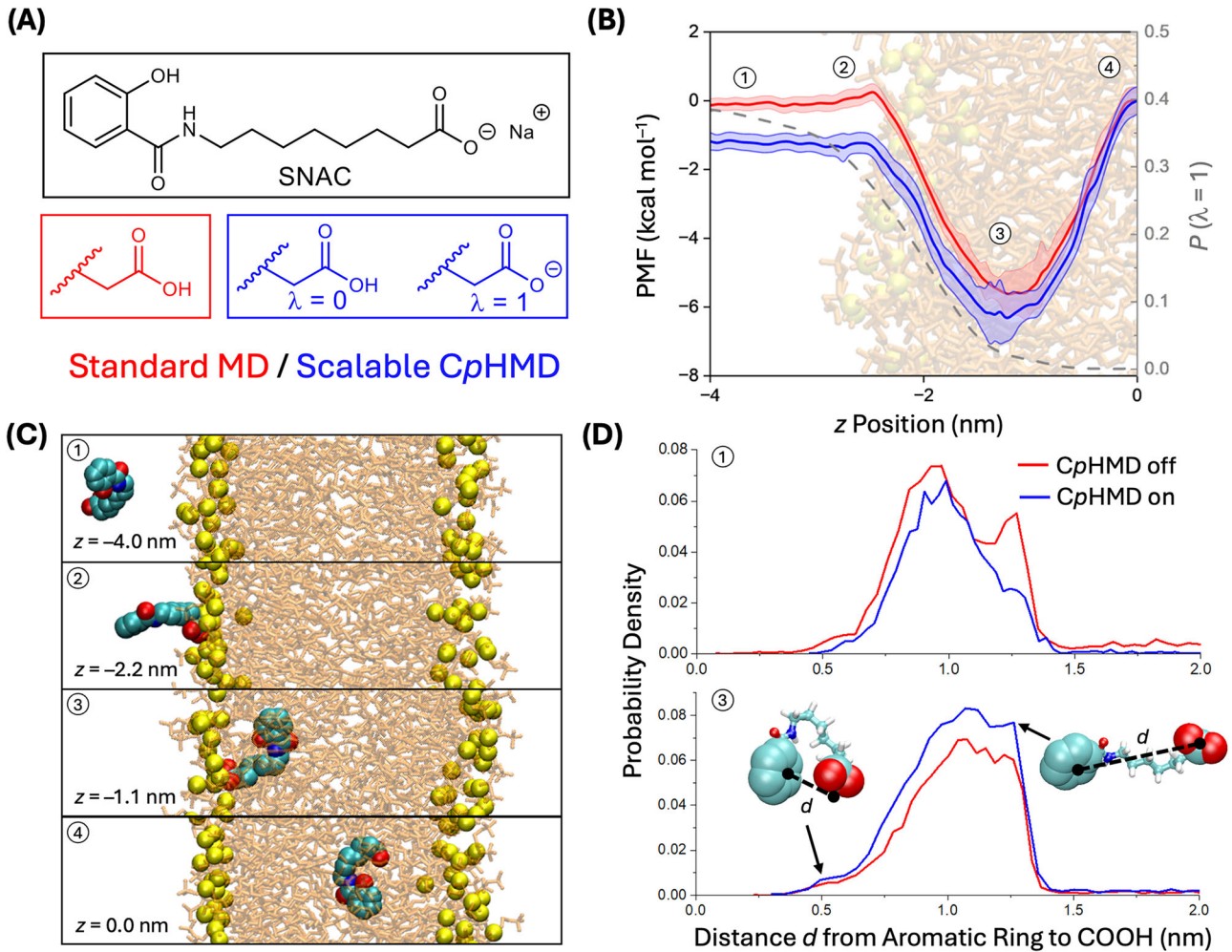

**Fig. 2 | Impact of dynamic protonation on the membrane insertion of the permeation enhancer SNAC. A** λ-Coordinate definition for the C$p$HMD model. **B** PMF profiles (310.15 K, 0.15 M NaCl, $pH$ = 5.0) for SNAC insertion into a POPC model membrane (64 lipids per leaflet). Data are provided as mean values ± SEM. The PMF compares a standard, fixed-protonation-state model for protonated SNAC (red curve) to the C$p$HMD model of SNAC (blue curve). The free energy curve was obtained by umbrella sampling as detailed in Supplementary Figs. 17 and 18, followed by standard WHAM analysis[70]. See Supplementary Video 1 for the trajectory of the corresponding pulling simulation, which generated the initial frames for the WHAM windows. Shaded bands represent error estimates obtained with bootstrapping ($n$ = 200) analysis implemented in the GROMACS WHAM analysis software. The distribution of lambda states at different z coordinates is shown with a grey dotted line to highlight the environment-dependent nature of SNAC protonation. These z-distance-dependent protonation probabilities were obtained from the C$p$HMD simulations by binning structures from select WHAM windows based on their z-positions and dividing the number of deprotonated frames by the total number of frames in each bin. Additionally, representative structures at key positions, both inside and outside of the membrane, are numbered and shown in panel (**C**). To evaluate how effectively different conformations of SNAC are being sampled using C$p$HMD as compared to standard methods, the distances $d$ between the aromatic ring (centroid of the aromatic carbons) and carboxylic acid (centroid of the COOH functional group) of SNAC were plotted as normalized histograms. **D** Superimposed histogram plots with C$p$HMD on (blue) and off (red) for SNAC outside (1) and inside (3) of the membrane show that both methods readily sample extended and contracted conformations of SNAC inside and outside the membrane. Representative coiled and linear structures of SNAC are included with arrows pointing to their distance on each graph. SNAC favors a more coiled structure inside the membrane as compared to outside the membrane with both methods. The trajectories of both umbrella sampling windows are shown in Supplementary Video 16 and Supplementary Video 17. Source data are provided as a Source Data file.

## Free energy profiles of SNAC entering/exiting a membrane

Figure 2B compares the free energy profiles for membrane permeation of SNAC with two different protonation state models: (1) A classical fixed protonation state force field, and (2) the C$p$HMD-based model. Both models show SNAC spontaneously incorporating into the membrane, in agreement with prior results obtained with fixed protonation state models[22]. However, with the C$p$HMD methodology, we now find that SNAC dynamically ionizes in the aqueous phase and then neutralizes to pass through the lipid bilayer membrane (see also Supplementary Fig. 19, as well as Supplementary Video 1). This process lowers the free energy for SNAC in the aqueous phase by ~1 kcal/mol, compared to the aqueous free energy obtained with the corresponding fixed protonation-state force field for protonated SNAC (see the

Supplementary Methods section as well as Supplementary Figs. 16–20 as well as Supplementary Video 16 and Supplementary Video 17 for details regarding the calculation of the potential of mean force (PMF) with umbrella sampling). The less biased distribution of SNAC between the water and membrane layer obtained with the C$p$HMD model revealed that SNAC can perform a dual role in monomerizing the peptide and facilitating permeation as described below.

To demonstrate that the C$p$HMD model applied in this work can reproduce the $p$Ka shifts of carboxylic acids in membrane-like environments, we applied the C$p$HMD model to a well-established oleic-acid model system. Self-assembled oleic acid demonstrates $pH$-dependent structures and composition-dependent apparent $p$K$_a$ values[35], with the apparent $p$K$_a$ value of oleic acid/glycerol monooleate systems ($p$K$_a \geq 6$)

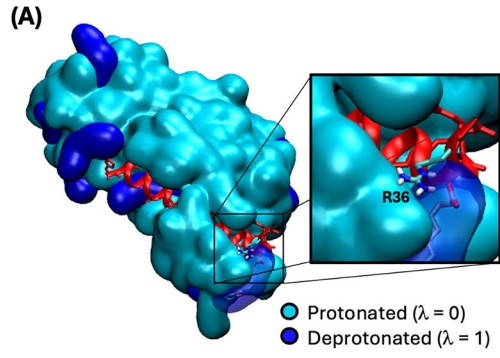

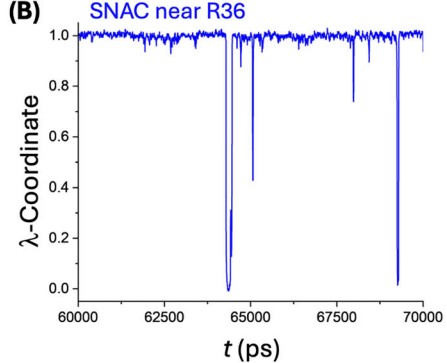

**Fig. 3 | Impact of dynamic protonation on the membrane insertion of the permeation enhancer SNAC. A** Self-assembly of 50 SNAC molecules (all treated with the CpHMD model) with semaglutide in aqueous solution after 50 ns (cyan: protonated SNAC ($\lambda < 0.5$); blue: deprotonated SNAC ($\lambda > 0.5$)). See Supplementary Video 4 for the trajectory of this self-assembly process. The inset highlights an ionic interaction between a deprotonated SNAC molecule and R36 of semaglutide, which illustrates how the peptide environment affects the pKa and with it the dominant protonation state of SNAC. **B** Plot of the $\lambda$-coordinate for the SNAC molecule that forms the salt bridge with R36. This ionic interaction stabilizes the deprotonated ($\lambda = 1$) state for SNAC resulting in a local $pK_a$ of -3.4 (calculated based on the Henderson−Hasselbach equation). Source data are provided as a Source Data file.

generally increasing with increasing oleic acid concentration. This experimentally-demonstrated[35] $pK_a$ trend (with oleic acid in micelles being significantly less acidic than oleic acid in its monomeric form ($pK_a = 4.8$)) thus represents a well-established model system, which enabled us to validate the CpHMD model for carboxylic acids structurally related to SNAC in a membrane-like environment with experimental $pK_a$ data. Overall, we found that performing a theoretical titration (Supplementary Fig. 31) of oleic acid−with $\partial V/\partial \lambda$ CpHMD coefficients obtained from simulations of monomeric oleic acid in water with 0.15 NaCl (see also Supplementary Figs. 14 and 15)−reproduces the experimentally observed shift in $pK_a$ value reasonably well. Specifically, 10 molecules of oleic acid were randomly assembled in a 7.5 nm³ box and simulated for 100 ns ($n = 3$) at different pH values to create the theoretical titration curve shown in Supplementary Fig. 31 ($pK_a^{theo} = 5.8$, fit with a Hendersson−Hasselbach model), which matches with the experimentally observed apparent $pK_a$ value of oleic acid/ glycerol monooleate systems ($pK_a = 6.0$)[35]. Overall, these results serve as proof of concept that the CpHMD methodology applied in this work can capture the influence of membrane-like environments on the $pK_a$ of hydrophobic carboxylic acids embedded in a membrane.

## Co-aggregation of SNAC and semaglutide in the aqueous phase

Next, with a larger CpHMD system containing 50 SNACs in a water box, we found that SNAC rapidly aggregates by itself and around semaglutide (Fig. 3 as well as Supplementary Video 2 and Supplementary Video 5) in water. These simulations also demonstrated that the protonation states of SNAC and semaglutide during co-aggregation are strongly influenced by the interactions between SNAC and semaglutide, as well as interactions with other SNAC molecules. This result

further highlights the need for a CpHMD model to capture these dynamic, environment-dependent effects. For instance, a salt bridge was formed between a SNAC molecule and R36 in semaglutide, which stabilizes the deprotonated state of the corresponding SNAC, resulting in a significantly lower local $pK_a$ of 3.4 (Fig. 3) of the corresponding SNAC molecule.

To experimentally validate the computationally observed co-association between SNAC and semaglutide, we conducted convection-corrected ¹H DOSY NMR experiments with buffered solutions of SNAC, semaglutide, and a 40:1 mixture of the two compounds. Our results (Table 1 and Fig. 4A) indicate that the viscosity-corrected hydrodynamic radii for SNAC and semaglutide increase for both compounds when mixed together. This result is consistent with the computationally observed co-aggregation of the two compounds. Notably, the ¹H DOSY NMR signals for SNAC exhibit (Fig. 4A and Table 1) two diffusion coefficients, suggesting that SNAC aggregates of at least two different sizes are formed. Overall, these combined experimental and computational observations provide a direct indication for aggregation between SNAC and semaglutide in the aqueous layer to (1) help monomerize the semaglutide in aqueous solution[10] and (2) improve the affinity of the semaglutide-SNAC aggregates to membrane-bound SNAC aggregates (which we hypothesize helps recruit the semaglutide to the membrane surface as discussed in detail below). See Supplementary Videos 11, 12, 13, 14, and 15 for visual depictions of this process.

## Dynamic SNAC aggregation in nonpolar environments

Individually, SNAC aggregation in aqueous media has been reported[9,36]. However, SNAC's behavior in non-polar environments is still mostly unknown, yet crucial, for facilitating the membrane permeation of semaglutide. Our observations now reveal that protonated SNAC also has the potential to aggregate in non-polar environments, as demonstrated by DLS (Fig. 4B, Supplementary Fig. 4) and through CpHMD simulations. Specifically, the CpHMD simulations show SNAC forming dynamic aggregates in $CH_2Cl_2$ (Fig. 5A, Supplementary Figs. 26−27, Supplementary Videos 3, and 4), in CTAB micelles (Fig. 5B, Supplementary Videos 7, 8, and 9), and inside a model membrane (Fig. 5D, Supplementary Videos 11, 12, 13, 14, and 15). The computational model indicates that SNAC aggregation in non-polar environments is driven by π-stacking between the aromatic SNAC residues (Fig. 5B) and hydrogen bonding, including branched hydrogen bonds that can lead to dynamic network formation inside the membrane (Supplementary Figs. 33−34).

## Table 1 | Evidence of SNAC/semaglutide co-aggregation in water based on convection-corrected ¹H DOSY NMR

| Sample | $D$ (m² sec⁻¹) | $R$ (nm) |
|---|---|---|
| 100 mM SNAC | $3.38 \times 10^{-10}$ | 0.74 |
| 2.5 mM semaglutide | $8.34 \times 10^{-11}$ | 3.41 |
| 100 mM SNAC + 2.5 mM semaglutide | SNAC: $2.79 \times 10^{-10}$ | 0.83 |
|  | SNAC:[a] $2.44 \times 10^{-10}$ | 0.94 |
|  | Semaglutide:[a] $6.17 \times 10^{-11}$ | 3.72 |

The table lists the diffusion coefficients ($D$) and corresponding hydrodynamic radii ($R$) of SNAC and semaglutide, calculated with the Stokes-Einstein Equation. The diffusion coefficient of the solvent was used to correct for slight variations in sample viscosity.
[a]We hypothesize that free SNAC and SNAC in the cluster around semaglutide exchange rapidly on the NMR timescale, which leads to distinct diffusion bands for SNAC and semaglutide resonances.

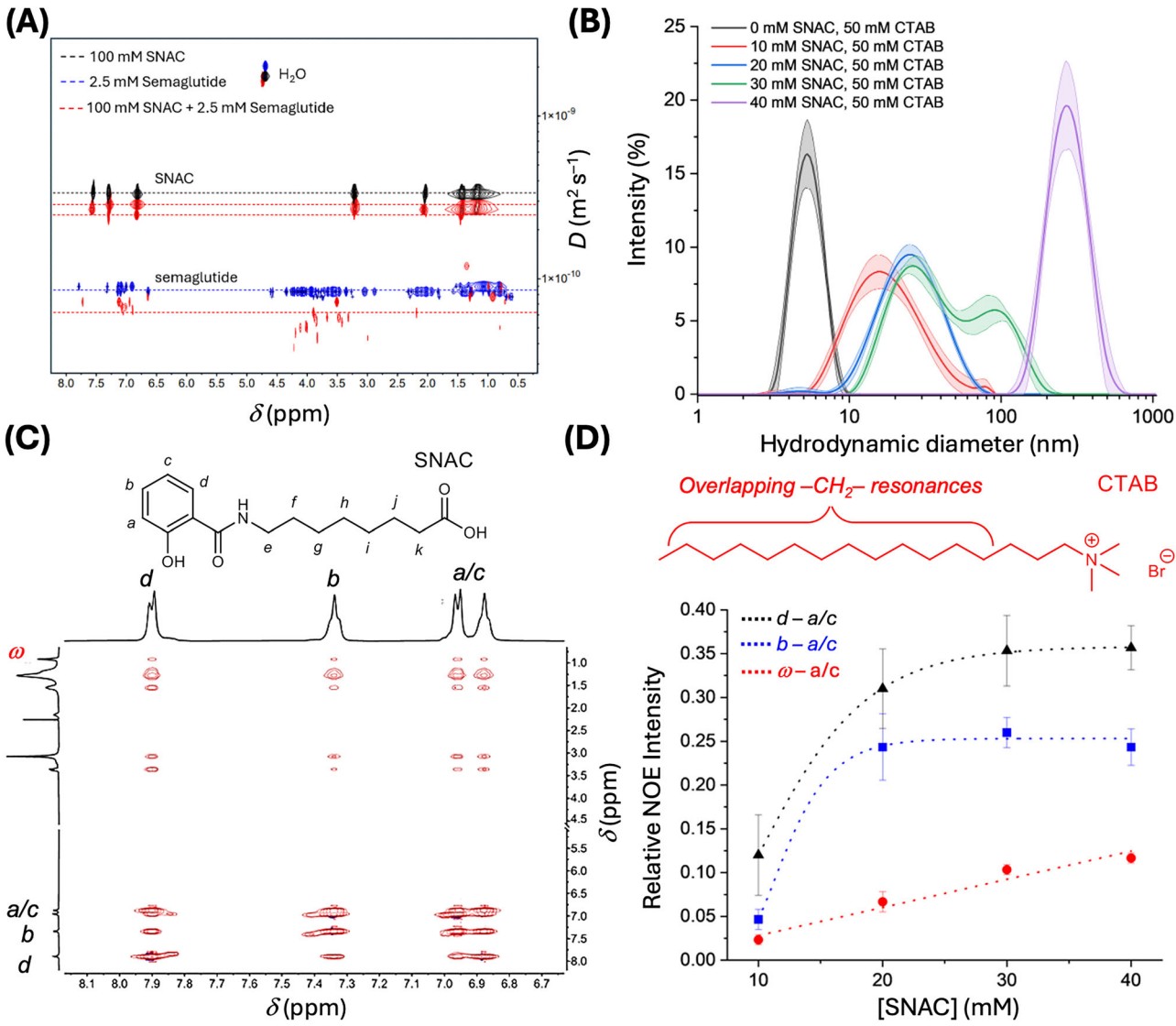

**Fig. 4 | Experimental evidence of SNAC aggregation in polar and non-polar environments.** **A** Stacked convection-compensated $^1H$ DOSY NMR spectra (800 MHz, 298 K, $D_2O$, $pH = 7.4$ in PBS Buffer, pulse sequence: dstebpg3s) of 100 mM SNAC (black), 2.5 mM semaglutide (blue), and a mixture of 100 mM SNAC and 2.5 mM semaglutide (red). Diffusion coefficients and hydrodynamic radii are detailed in Table 1. **B** DLS spectra (298 K, mean intensity ± standard deviation, $n = 10$) of SNAC in 50 mM solutions of CTAB demonstrating incorporation of SNAC into surfactant micelles. Additional DLS spectra of the conjugate acid of SNAC aggregating in non-polar solvents ($CDCl_3$) are provided in Supplementary Fig. 4.

**C** $^1H$-$^1H$ NOESY NMR (500 mHz, 298 K, $D_2O$, $pH = 5.6$, pulse sequence: *noesygpph*) spectra of 40 mM SNAC and 50 mM CTAB with highlighted crosspeak signals to indicate the incorporation of SNAC in the CTAB micelle. **D** Crosspeak intensities of SNAC ($n = 3$, labeled in Fig. 4C and referenced to the ω CTAB diagonal intensity) are concentration dependent. Solutions for the NOESY and DLS titration experiments were buffered with SNAC at $pH = 5.6$ by using a 4:1 mixture of SNAC and its conjugate acid. Data are provided as mean values ± SD. Source data are provided as a Source Data file.

Experimental evidence for intermolecular hydrogen bonding between SNACs was obtained through a $^1H$ NMR titration of protonated SNAC in $CDCl_3$ (Supplementary Figs. 1–2). $CDCl_3$ was chosen as a model system since it is a well-established surrogate for the nonpolar environment of lipid bilayer membranes[8,24–26,37]. From these NMR titrations (Supplementary Fig. 1), we found that the amide and carboxylic acid resonances of SNAC are highly sensitive to sample concentration, as expected for intermolecular hydrogen bond formation. Specifically, the -0.1 ppm downfield shift of the amide resonance with increasing SNAC concentration is attributed to an increase in hydrogen bonding, which deshields the amide and carboxylic acid peaks[38]. Furthermore, at higher SNAC concentrations (>70 mM), the titration curve deviates significantly from a simple 1:1 binding model (Supplementary Fig. 2), indicating the formation of larger aggregates, which were also observed by DLS (Supplementary Fig. 4). Additionally,

convection-corrected $^1H$ DOSY NMR spectra show that the diffusion coefficient of SNAC in $CDCl_3$ decreases with increasing SNAC concentration (Supplementary Fig. 3), which is also consistent with SNAC aggregate formation. Based on the measured diffusion coefficients and assuming spherical aggregates, the average aggregate volumes for 5 mM and 100 mM samples of protonated SNAC in $CDCl_3$ (298 K) were calculated to be 0.42 $nm^3$ and 1.05 $nm^3$, respectively, using the Stokes-Einstein relationship. The diffusion coefficient of the solvent peak remained nearly identical for the two SNAC solutions in $CDCl_3$ ($2.68 \times 10^{-9}$ $m^2 s^{-1}$ for the 5 mM solution and $2.71 \times 10^{-9}$ $m^2 s^{-1}$ for the 100 mM solution), ruling out significant viscosity effects on the apparent diffusion coefficients of SNAC at different concentrations. Even larger SNAC aggregates (-150 nm diameter) were observed by DLS in $CDCl_3$ (Supplementary Fig. 4), which is more sensitive to larger particles than NMR[39].

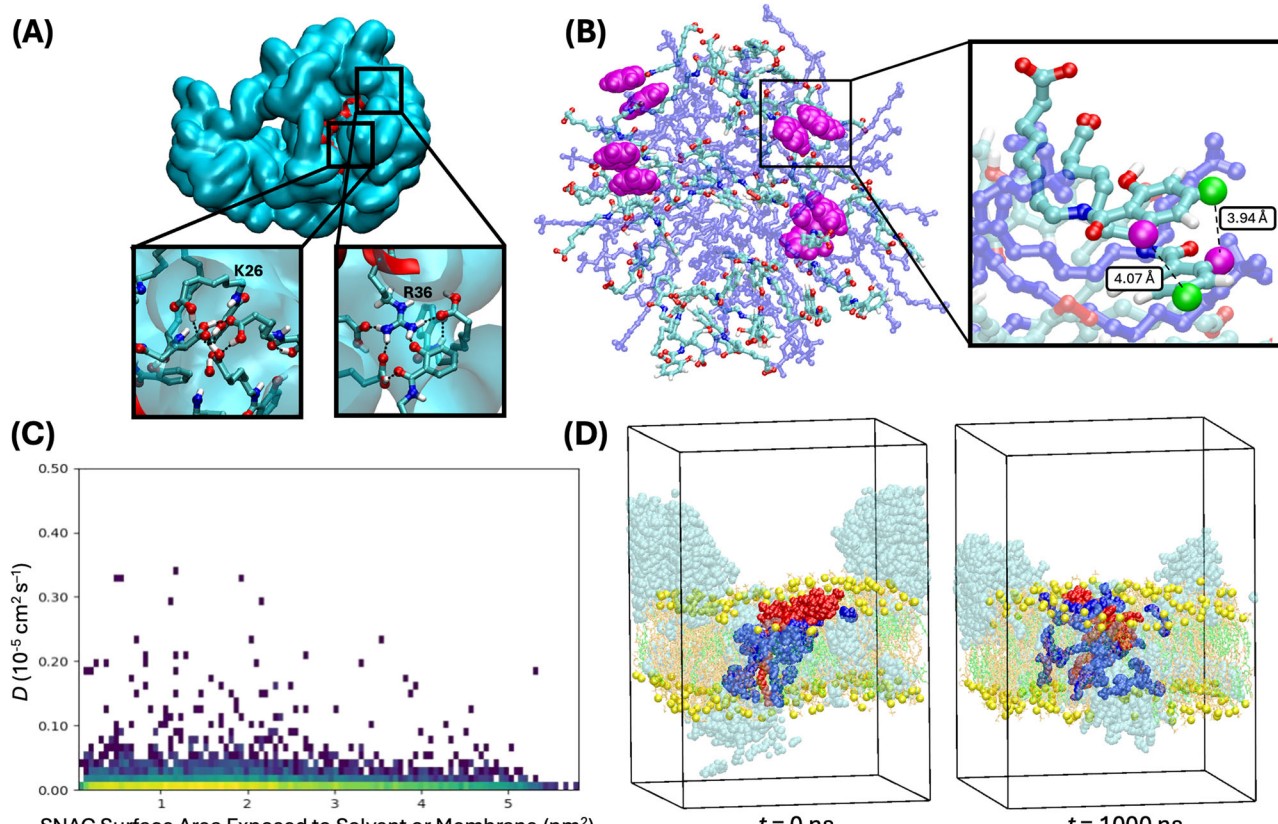

**Fig. 5 | Computational support for dynamic SNAC aggregation in non-polar environments. A** Self-assembly of 50 SNAC molecules with semaglutide in $CH_2Cl_2$ after 50 ns (cyan: protonated SNAC with λ < 0.5; blue: deprotonated SNAC with λ > 0.5). The insets highlight stabilizing interactions between SNAC and titratable peptide residues. See Supplementary Fig. 27B for the corresponding aggregation of just SNAC in $CH_2Cl_2$, as well as Supplementary Video 3 and Supplementary Video 4 for the corresponding trajectories. **B** C*p*HMD simulation of a SNAC/CTAB micelle highlighting π-stacking interactions of aggregated SNAC molecules in the micelles that are also observed (Fig. 4C/D) in $^1H$-$^1H$ NOESY NMR experiments. SNAC molecules are shown with carbons in opaque cyan, polar hydrogens in white, nitrogens in dark blue, and oxygens in red. CTAB is shown in blue in semi-transparent mode. Key hydrogen atoms observed in the $^1H$-$^1H$ NOESY NMR experiments are highlighted in space-filling mode with H-*d* shown in magenta, and *H*-*b* in light green. **C** Relationship between the exposed surface area of SNAC molecules (which

correlates with the position of SNAC inside the membrane cluster) and the calculated diffusion coefficients *D* of the corresponding SNAC molecules across the 1 μs membrane simulations (*n* = 4). This plot clearly shows that mobile SNAC molecules exist both inside and outside the SNAC aggregates in the membrane. **D** Snapshots of the initial and final frames of the 1 μs membrane simulation (first replica) with all SNAC molecules initially present within 7.5 Å of the semaglutide (red) highlighted in dark blue. These snapshots indicate that the SNAC molecules in the SNAC cluster disperse throughout the SNAC cluster during the simulation, which shows that these SNACs are mobile. See Supplementary Fig. 35B−D for analogous plots of the other three simulation replicas. Additional examples with other dynamic SNAC molecules that are part of a membrane cluster are shown in Supplementary Video 12, Supplementary Video 13, and Supplementary Video 14. Source data are provided as a Source Data file.

Finally, the aggregation of SNAC in a membrane-like environment was also demonstrated using a detergent micelle model (CTAB), as evidenced by both DLS and NMR experiments (Fig. 4B−D). From these experiments, we found that the size of the SNAC/CTAB aggregates observed with DLS is sensitive to the SNAC concentration, resulting in aggregates of increasing size and complexity at higher SNAC concentrations. At the same time, $^1H$ NMR titrations of SNAC into solutions of CTAB (Supplementary Figs. 28 and 29) demonstrate significant broadening of SNAC and CTAB resonances, which also supports their aggregation into larger structures. Furthermore, $^1H$-$^1H$ NOESY NMR experiments exhibited concentration-dependent SNAC cross-peaks. Specifically, we found the aromatic SNAC/SNAC NOE coupling intensities to be sensitive to the increasing SNAC concentration, suggesting that SNAC aggregates in the CTAB micelle environment, likely via a combination of π-stacking between the aromatic SNAC residues and hydrogen bonding interactions. This finding is also supported by C*p*HMD simulations of SNAC with CTAB micelles (Fig. 5B, Supplementary Fig. 30, Supplementary Videos 7, 8, and 9), which clearly show π-stacking between the corresponding $^1H$ NMR signals, for which the NOE cross peaks initially increase rapidly and non-linearly with

increasing SNAC concentration, consistent with SNAC aggregation in/ around the micelles. In turn, if SNAC were getting evenly distributed in/ around the CTAB micelles (i.e., not aggregating in/on the micelles), the NOE cross-peaks between the aromatic SNAC residues would be expected to increase at a similar rate as the CTAB/SNAC NOE cross-peaks. In contrast, the relative CTAB/SNAC NOE coupling intensity of the very clearly defined ω-a/c crosspeak was found to increase only very slowly and in a primarily linear fashion with increasing SNAC concentration (Fig. 4D). We believe that this relatively small increase in CTAB/SNAC NOE cross-peak intensity is caused by the naturally more intense SNAC $^1H$ NMR signals arising with increasing SNAC concentration. Moreover, the relatively small change in NOE intensity observed for the CTAB/SNAC ω-a/c NOE crosspeak also indicates that aggregation size effects on the correlation times are very likely small compared to the changes in NOEs caused by SNAC aggregation in our system, consistent with the fact that the large CTAB/SNAC aggregates (Fig. 4B) all lie in the slow-tumbling regime for NOESY NMR (with all NOE cross peaks for SNAC and CTAB being negative, i.e., with the same sign as the diagonal peaks as clearly shown in Fig. 4C and Supplementary Fig. 28B). Overall, the experimentally observed aggregation of

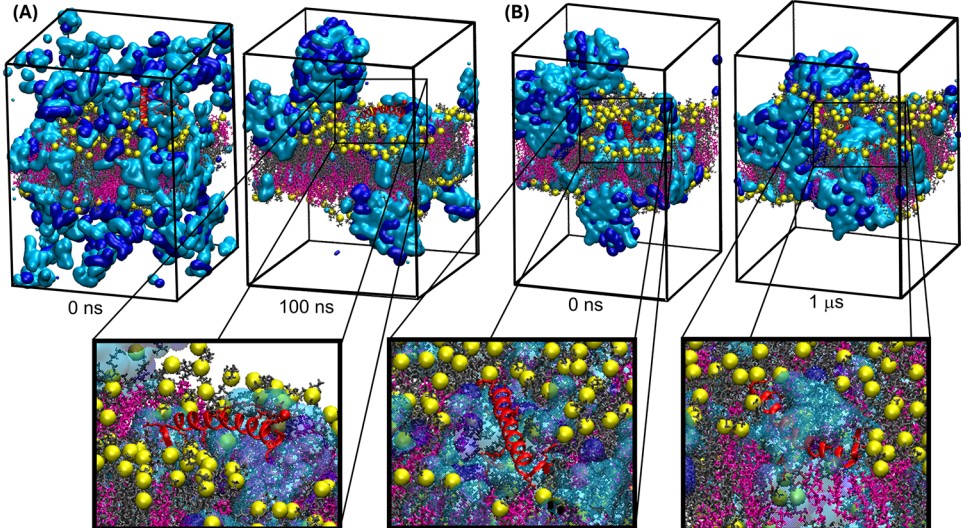

**Fig. 6 | Unbiased C*p*HMD Simulations showing semaglutide spontaneously adhering to and incorporating into the membrane in the presence of SNAC.** The model contained semaglutide (red), 60% POPC lipids (yellow/grey), 40% cholesterol (pink), and 400 SNAC molecules (cyan: protonated with λ < 0.5; blue: deprotonated with λ > 0.5). **A** 100-ns C*p*HMD simulation during which a SNAC-semaglutide aggregate spontaneously forms and adsorbs to the membrane surface. **B** 1-μs C*p*HMD simulation, which was started from a snapshot with semaglutide's fatty acid tail anchored in the membrane. After a few hundred nanoseconds semaglutide spontaneously starts to sink into the SNAC defects formed inside the membrane as the peptide drug is getting buried by SNAC molecules moving from the center of the membrane toward the semaglutide. Similar behavior was also observed in three additional replicas generated from different random seeds (including two replicas with a different initial starting conformation with the lipid tail buried less deep in the membrane). Movies of all trajectories are included as Supplementary Video 11, Supplementary Video 12, Supplementary Video 13, Supplementary Video 14, and Supplementary Video 15.

SNAC in non-polar environments supports our computational results, which show that dynamic SNAC aggregates can form in $CH_2Cl_2$, in detergent micelles, and in the interior of a lipid bilayer membrane. At the same time, the dynamic nature of the SNAC aggregates in the membrane is demonstrated in Fig. 5C/D, Supplementary Figs. 35, as well as in Supplementary Videos 12, 13, and 14, which all highlight the significant movement of selected SNAC molecules in the membrane (all of them part of SNAC membrane clusters) during the 1-μs-long C*p*HMD membrane simulations.

### Semaglutide incorporation into the membrane with SNAC

To directly observe the mechanism of peptide permeation in the presence of SNAC computationally, we performed unbiased C*p*HMD simulations with a system containing 400 SNAC molecules (periodic box size: 11 nm × 11 nm × 15 nm), semaglutide, and a model POPC/cholesterol epithelial membrane (see Fig. 6A). The composition of the system was based on the formulation of the Rybelsus® tablet[40] and the fact that the localized concentration of SNAC around the tablet is significantly higher than what would be expected after even distribution in the stomach, as the tablet is located directly on the stomach epithelia[41]. The concentrations of SNAC and semaglutide at the site where the tablet is located in the stomach have been measured at roughly 280 mM and 600 μM, respectively[9]. To match these relatively high concentrations localized under the tablet, semaglutide (800 μM) and SNAC (350 mM) were randomly placed around the membrane and the system was simulated for 100 ns to equilibrate. The membrane was constructed using Packmol[42], with its composition based on estimates of the cholesterol content in epithelial cells[43,44] and previously reported simulations of SNAC[45,46]. During the first 5 ns, semaglutide once again rapidly aggregated with neighboring SNAC molecules in the aqueous layer. Subsequently, semaglutide clusters in the water layer spontaneously fused with a membrane-bound SNAC aggregate, positioning (Supplementary Video 10) the peptide flat on the membrane surface after ~50 ns. Similar synergistic aggregation mechanisms have been described previously for antimicrobial peptides[47].

Next, given its hydrophobic nature, we hypothesized that the lipid tail of semaglutide (which includes[48,49] the fatty acid chain and γGlu-2xOEG linker ligated to K26 as shown in Supplementary Figs. 5 and 6) likely inserts first into the membrane, serving as a membrane anchor for the peptide. To confirm this hypothesis, we calculated the free energy profile for membrane insertion of the semaglutide lipid tail with the C*p*HMD method, using umbrella sampling followed by WHAM analysis[50] (Supplementary Figs. 21–24 and Supplementary Video 6). Our results showed that incorporating the lipid tail into the membrane center is favorable overall ($\Delta G$ = −5.0 kcal mol⁻¹). However, pulling the acid end of the lipid tail to the opposite leaflet imposed a 7.5 kcal mol⁻¹ free energy barrier due to the γGlu-2xOEG linker being pulled past the phosphate headgroups. Based on these results, we constructed an initial structure with the lipid tail of semaglutide anchored inside the membrane for extended, 1-μs unbiased C*p*HMD simulations of this system, as shown in Fig. 6B. This simulation was repeated with a different random seed, which yielded similar results and also showed the peptide sinking partially into the SNAC-filled membrane.

As this simulation progressed (see Supplementary Video 11 for a visual depiction of this process), the initial, smaller SNAC aggregates gradually expanded within the membrane, forming larger SNAC-filled membrane defects spanning the phospholipid head groups (Supplementary Fig. 25). Additional SNAC then continued to aggregate around the membrane-anchored peptide, which began to slowly sink into the membrane after ~100 ns. Throughout the remainder of this unbiased simulation, semaglutide continued to sink into the membrane as more SNAC molecules enveloped the hydrophobic portions of the peptide (see Fig. 6B and Supplementary Video 11). Based on these results, we hypothesize that—as semaglutide sinks through the SNAC membrane defects—there is a dynamic redistribution of the surrounding SNAC molecules. This hypothesis was also confirmed by (1) visualizing the dispersion of all the SNAC molecules initially surrounding the semaglutide during the simulation (Fig. 5D) and (2) by plotting the diffusion coefficients of all the SNAC molecules within the membrane SNAC clusters, which confirmed (Fig. 5C) that mobile SNAC molecules exist throughout the SNAC clusters in the membrane. The mobility of the

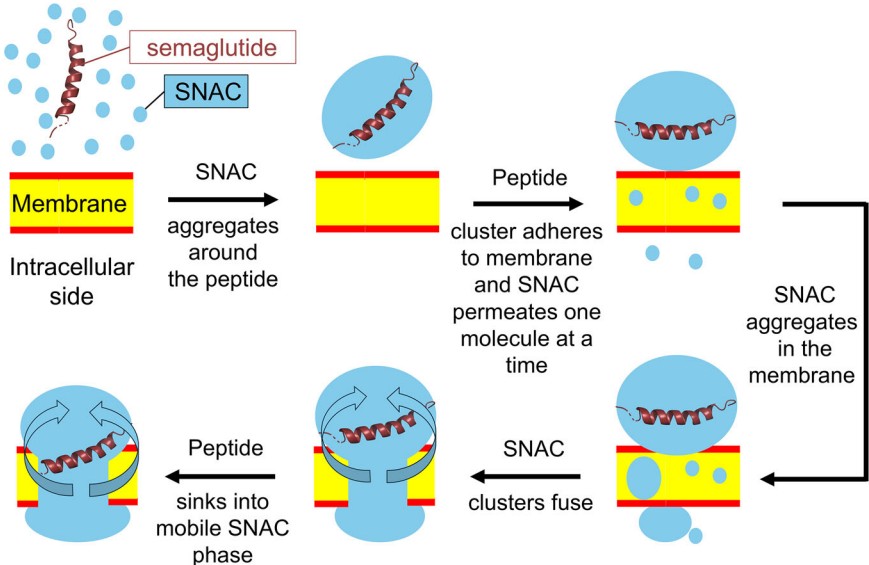

**Fig. 7 | A possible molecular mechanism for SNAC-assisted membrane permeation of semaglutide.** Curved arrows represent the movement of mobile SNAC molecules.

SNACs in the membrane clusters can then gradually lead to the peptide sinking into the membrane, as observed clearly in some of our unbiased simulations (see, e.g., Supplementary Video 11). Notably, while SNAC got incorporated into the membrane, the membrane remained stable throughout the simulations, with only minimal water getting incorporated (Supplementary Fig. 32). Then, as new SNACs are exposed to the exterior of the defects, they once again move to the top of the cluster, continuing the sinking process. This proposed mechanism, which resembles sinking into quicksand, is illustrated with curved arrows in Fig. 7.

## Discussion

With a combination of molecular dynamics simulations and experimental methods, we have now observed how a polar peptide like semaglutide can spontaneously sink into a membrane in the presence of a permeation enhancer (SNAC), to get delivered across the intestinal barrier. Dynamic protonation of weakly ionizable sites is crucial for this process, which we were able to observe with a C*p*HMD model that can accurately represent the environment- and aggregation-/conformation-dependent charges in this system. While dynamic protonation of ionizable sites is well-known to influence biological processes[30,31,33,51–55] and membrane permeability for small molecules (see Fig. 1A for a seminal example)[6,56–60], this work has now advanced the application of C*p*HMD models to permeation-enhancer facilitated peptide permeation by treating all the permeation enhancers and all ionizable sites on a peptide drug (> 400 ionizable groups in total) with a C*p*HMD model.

Furthermore, while both SNAC and semaglutide are known to form aggregates by themselves in buffered aqueous solutions based on DLS and DOSY NMR[36,61], our C*p*HMD simulations have clearly demonstrated co-association of the peptide with the permeation enhancer in the aqueous phase. We hypothesize that this observed co-aggregation is primarily responsible for the previously reported monomerization[10] of the peptide drug in an aqueous buffer in the presence of SNAC, a mechanism that was previously unknown in molecular detail[9]. In addition, since we found a clear affinity of SNAC toward itself and semaglutide in both aqueous and hydrophobic environments, the co-aggregation of SNAC with semaglutide might also play an important role in helping to recruit the SNAC-bound semaglutide from the aqueous layer to the membrane (with the SNAC around the semaglutide fusing with membrane-bound SNAC). Based on our combined computational and experimental results, we propose

the overall mechanism shown in Fig. 7 for peptide permeation with SNAC. Key steps include:

(1) Excess SNAC around the tablet forms aggregates with semaglutide and the membrane.
(2) SNAC molecules leave the aqueous cluster one at a time, incorporating into the membrane to form dynamic, SNAC-filled membrane defects, which allow semaglutide to bypass the barrier formed by the phospholipid head groups.
(3) Solution-based semaglutide-SNAC aggregates fuse with dynamic, membrane-bound SNAC-filled membrane defects, adsorbing semaglutide onto the membrane surface.
(4) The fatty acid chain of semaglutide then likely anchors the peptide to the membrane, with SNAC molecules in the membrane creating a "quicksand-like" effect, enabling the peptide to slowly sink into the membrane.
(5) The peptide keeps sinking through the dynamic SNAC-filled membrane defects until it reaches the other side of the membrane, where it then exits with excess SNAC, maintaining membrane integrity after peptide passage.

In summary, our C*p*HMD model revealed SNAC's dual role in peptide membrane permeation. First, in the water layer, SNAC associates with semaglutide, acting as a surfactant to help monomerize the peptide while remaining partially deprotonated. Second, upon membrane insertion, SNAC neutralizes and forms dynamic membrane defects to help facilitate membrane permeation of the peptide drug. Our proposed mechanism for the passive membrane permeation of semaglutide is substantiated by accurate C*p*HMD atomistic models (with all weakly ionizable functional groups in both the peptide and all the permeation enhancers modeled with C*p*HMD). While it delineates one of numerous potential pathways for semaglutide to traverse a membrane in the presence of SNAC[62,63], the integration of scalable atomistic C*p*HMD elevates it to one of the most expansive and accurate models documented to date. Moreover, the overarching principles elucidated in this study—specifically, SNAC's behavior in nonpolar environments, its dynamic aggregation with semaglutide in both polar and nonpolar contexts, and the formation of SNAC-filled dynamic defects within the membrane—present foundational concepts, which hold significant potential to inform and refine the development of new peptide drug/permeation enhancer combinations in the future. The adhesion of semaglutide to the membrane, which was observed in our

CpHMD simulations, also agrees with the initial steps for cell translocation of known cell-penetrating peptides[64]. Overall, this work serves as the basis for future models that will likely further refine this complex process, as our results provide a new, viable mechanism for cell permeation, helping to advance the field of oral peptide drug delivery, including potentially for macrocyclic peptides in the future[8,65–67].

## Methods

### CpHMD Simulations

The simulations described in this work utilized a scalable CpHMD method implemented by the Groenhof and Hess groups in a custom fork of GROMACS 2021[29]. Due to the environment and conformation-dependent $pK_a$ values of all the weakly ionizable functional groups in the peptide and the SNACs, the enhanced accuracy of a CpHMD model is crucial to meaningfully predict the charges and interactions between semaglutide, SNAC, and the membrane[32]. Specifically, we found that the charges of the permeation enhancers and the peptides (which are directly controlled by the protonation states) are environment-dependent, which significantly affects the free energy change associated with membrane insertion of SNAC (Fig. 2B). In this work, the CpHMD model (see the Supplementary Methods section as well as Supplementary Figs. 7–15 for details regarding the parametrization) enabled the simulation of all the titratable sites in the peptide and ~400 permeation enhancers under constant pH conditions at ~80% of the efficiency of standard MD simulations[29]. The 400:1 molar ratio of SNAC:semaglutide in the simulations was chosen based on the current formulations in Rybelsus*[40], which contains ~1 mmol of SNAC per tablet in a ~380:1 to ~760:1 SNAC:semaglutide molar ratio. Simulations of SNAC and CTAB were performed as follows: First, a CTAB micelle structure was generated by randomly inserting 25 molecules of CTAB (50 mM) into a periodic boundary box, which was then solvated and equilibrated for 100 ns. Convergence for all the CpHMD equilibration runs (see the Supplementary Methods section as well as Supplementary Tables 1 and 2 for additional computational details) was verified with plots of the total energy vs. time (Supplementary Figs. 36–49). Second, 20 SNAC molecules (which corresponds to an overall SNAC concentration of 40 mM) were randomly added around the pre-equilibrated CTAB micelle in a new box, and the new system was again solvated and equilibrated for 100 ns with CpHMD, which led to self-assembly of a mixed SNAC/CTAB micelle. Lastly, four preequilibrated SNAC/CTAB micelles were placed in a box (corresponding to an overall concentration of 100 mM CTAB and 80 mM SNAC) and then solvated and equilibrated with CpHMD for 100 ns, resulting in a larger SNAC/CTAB aggregate.

To enable these large-scale CpHMD simulations, we first generated CpHMD parameters (reported in Supplementary Table 1) for the titratable groups of SNAC and non-standard amino acids following standard best practices as detailed in the Supplemental Methods section[34]. All of our CpHMD parameters are provided in Supplementary Table 1 and Supplementary Figs. 8–15. Furthermore, to mimic the localized SNAC-buffered environment around the tablet, which is less acidic than the rest of the stomach[9], the CpHMD simulations were performed at $pH = 5.0$ (the $pK_a$ value of SNAC in water)[68]. Additional details regarding all the CpHMD parametrizations and simulations performed are provided in the Supplementary Methods section and in Supplementary Tables 1 and 2.

### $^1$H NMR Titrations of SNAC

SNAC was dissolved in 0.7 mL CDCl$_3$ containing 1% (v/v) TMS at concentrations ranging from 5 mM to 100 mM with gentle heating. Spectra were recorded immediately after the samples were dissolved on a Bruker NEO500-2 spectrometer for 32 scans at 298 K, and all resonances were referenced to TMS as the internal standard. Clear concentration-dependent shifts (Supplementary Fig. 1) of the amide (N-H) resonance in SNAC at ~6.3 ppm were observed, which is indicative of attractive supramolecular interactions dominated by hydrogen bonding (shown in Supplementary Figs. 33 and 34) between the SNAC molecules in CDCl$_3$ solution. The resulting $^1$H NMR chemical shift-concentration plots (Supplementary Fig. 2) were then analyzed with the Dynafit[69] (v 4.11.110) software package using a simple 1:1 homodimerization binding model for SNAC concentrations <70 mM. However, for larger SNAC concentrations, deviations from a simple 1:1 binding model were observed, which is consistent with the formation of larger, dynamic SNAC aggregates that are also observed in DLS (Fig. 4B) and by all-atom molecular dynamics simulations (Fig. 5 and Supplementary Figs. 26, 27, and 35).

### $^1$H NMR Titrations of SNAC with 50 mM CTAB

Samples were prepared by creating a stock solution of SNAC (36 mg, 120 mmol) and its conjugate acid (8.4 mg, 30 mM) in MeOH (2.0 mL), and a solution of CTAB (91 mg, 250 mmol) and sodium trimethylsilylpropanesulfonate (DSS) (2.2 mg, 10 mmol) in D$_2$O (5.0 mL). SNAC solutions were then aliquoted into 10 mL scintillation vials and evaporated to dryness with a rotovap to deposit thin films of 40, 30, 20, and 10 mmol of SNAC, respectively. To the films, 1.0 mL of the CTAB solution in D$_2$O was added, and the mixture was sonicated for 5 min before being gently heated, resulting in clear solutions. Solutions were then transferred to NMR tubes and spectra recorded using a Bruker NEO500-2 spectrometer for 32 scans at 298 K, and all resonances were referenced to DSS as the internal standard. All experiments were performed in triplicate with fresh solutions prepared for each replica.

### $^1$H Diffusion-ordered spectroscopy (DOSY) NMR

The NMR samples for $^1$H DOSY NMR spectroscopy were prepared in 0.3 mL CDCl$_3$ or D$_2$O at 298 K and recorded with solvent-matched Shigemi NMR tubes (Wilmad CMS-005B for CDCl$_3$ and BMS-005B for D$_2$O) to minimize convection. The caps of the NMR tubes were sealed with parafilm before being lowered into the magnet. The $^1$H DOSY NMR spectra were acquired on a Bruker Avance-III-800 (800 MHz) spectrometer with a standard-bore Bruker ultra stabilized magnet, equipped with a 5 mm QCI Z-gradient cryoprobe containing a cold $^{13}$C preamplifier, a Z-axis field gradient module, four RF channels with waveform generation, H2-decoupling capability, a sample temperature control unit, and a Linux host computer running TopSpin 4.1.1. The DOSY pulse program used was a standard double-stimulated-echo experiment with bipolar gradient pulses and convection compensation (Pulse Sequence: dstebpg3s implemented in TopSpin). A total of 16 different spectra were recorded for all samples using a 98% pulse gradient for the samples in CDCl$_3$ and a 69% pulse gradient for the samples in D$_2$O. The data were processed in MestReNova (v 14.3.3) using the Peak Fit method. The hydrodynamic radii were estimated using the Stokes-Einstein equation. This equation was solved for $R$ (the solvodynamic radius) using appropriate values for the solvent viscosity $\eta$ from the literature.

### $^1$H-$^1$H NOESY NMR Spectroscopy

The same samples used for the SNAC titration with 50 mM CTAB were also used for the NOESY experiments. Spectra were collected on a Bruker NEO500-2 spectrometer using the *noesygpph* pulse sequence (with a mixing time of 250 ms). Data were processed using MestReNova (version 14.3.3), and the corresponding $^1$H NMR spectra were used as the reference for external projections in 2D plots.

### Dynamic light scattering of SNAC in CDCl$_3$

Samples were prepared by dissolving protonated SNAC in CDCl$_3$ (Cambridge Isotopes Cat. No.: DLM-29-0) in 15 mL scintillation vials with gentle heating. The dissolved samples were then filtered through celite and transferred to an ultra-low volume quartz cuvette (ZEN2112) for the acquisition of the DLS spectra. The concentration of the

samples was quantified with [1]H NMR after filtration through celite. Quantitation was performed with 1,3,5-tribromobenzene as the internal standard and the *D*1 NMR acquisition delay parameter in Topspin was set to 10 s to ensure accurate NMR integrations. For DLS spectrum acquisition, each DLS spectrum was recorded 10 times. Samples were given 15 s to equilibrate in between the different measurements.

### Dynamic light scattering of SNAC with CTAB micelles

Samples were prepared by creating a stock solution of SNAC (36 mg, 120 mmol) and its conjugate acid (8.4 mg, 30 mM) in MeOH (2.0 mL), and a solution of CTAB (91 mg, 250 mmol) and sodium trimethylsilyl-propanesulfonate (DSS) (2.2 mg, 10 mmol) in $D_2O$ (5.0 mL). SNAC solutions were aliquoted into 10 mL scintillation vials and evaporated to dryness with a rotary evaporator to deposit thin films of 40, 30, 20, and 10 mmol of SNAC, respectively. To the films, 1.0 mL of the CTAB solution in DI water was added, and the mixture was sonicated for 5 min. It was then gently heated and cooled again to room temperature to ensure clear solutions before data collection. Data were collected using a low-volume (50 μL, model number: ZEN2112) quartz cuvette. Each spectrum was recorded 10 times, with 15 s provided for equilibration between measurements.

### Reporting summary

Further information on research design is available in the Nature Portfolio Reporting Summary linked to this article.

## Data availability

Additional data supporting the findings of this study are provided in the Supplementary Data files, including input and output coordinate files from the molecular dynamics (MD) simulations (Supplementary Data 1) as well as movies of selected MD trajectories (Supplementary Video 1 – Supplementary Video 17). Source data for all graphs are supplied with this paper as a Source Data file. The complete MD trajectories, which amount to several hundred gigabytes, exceed the size limits of currently accessible data repositories. Therefore, the full trajectory files are available from the corresponding author upon request. Source data are provided with this paper.

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

## Acknowledgements

We thank Dr. Huaping Mo for assistance with DOSY NMR spectroscopy, Dr. Jianing Li for helpful discussions, the Purdue Rosen Center for Advanced Computing for providing computational facilities, and the Purdue Interdepartmental NMR Facility for providing access to NMR spectrometers used to carry out this research. We thank the NIH for funding this work under a MIRA grant awarded to S.T.S. (1R35GM147579). Part of the computational resources were also supported by an NSF CAREER Award (CHE-1848444/2317652).

## Author contributions

S.T.S. guided the project, discussed the experimental and computational results, and helped write the paper. K.J.C. performed and analyzed the simulations and part of the experiments reported in the paper. K.T.F. performed and analyzed part of the experiments reported in the paper. All the authors discussed the results and revised the paper.

## Competing interests

The authors declare no competing interests.
