## [Transparent Peer Review file · Nature Communications]

Permeation Enhancer-Induced Membrane Defects Assist the Oral Absorption of Peptide Drugs

Corresponding Author: Professor Severin Schneebeli

Version 0:

Reviewer comments:

Reviewer #1

(Remarks to the Author)

Using a constant-pH molecular dynamics (cpHMD) algorithm and experimental measurements, this work reports a novel membrane permeation mechanism for a relatively large polar peptide (semaglutide) paired with a permeation enhancer salcaprozate sodium (SNAC). Computation of membrane insertion and permeation of peptides that contain charged side chains amino acids is a difficult problem involving mainly changes of local charges, membrane distortions, conformational changes of the peptide, translational and orientational degrees of freedom of the peptide and possibly other reaction coordinates. This work emphasizes on the evaluation of the protonation states of titratable groups on both semaglutide and the SNAC molecules and the membrane perturbations caused by the large amount of SNAC used in the oral pharmaceutical formulation of semaglutide. In exploring those two aspects of the problem, this work is novel and provides interesting insights that could be used to study permeation of other charged systems (peptides, drugs, oligonucleotides, etc.) where the presence of permeation enhancers is required to enable membrane translocation.

I don't have a working expertise to comment on the experimental measurements done by the authors. In the following paragraphs I will itemize some comments about the simulation part.

- (1) The simulation protocols followed in the work are adequate. The cpHMD tool added to Gromacs is a very recent development that have been used mostly to study protonation states of peptides and proteins in aqueous environments. I wonder if the protocols described by the designers of the phbuilder package to parametrize the coefficients for pH control work correctly in a membrane environment given that the runs used for parametrization of the dVdl coefficients contains the charge groups inside a water box. Maybe the authors can comment on that possible issue.
- (2) Simulations to compute Potential of Mean Force for insertion of SNAC or the semaglutide tail with umbrella sampling used membrane patches of 32 POPC phospholipid molecules per leaflet. The lateral box size is probably small and that could produce some simulation artifacts. Why did the authors use that small membrane patches to study permeation of those two relatively long molecules?
- (3) In the simulations that contain semaglutide, SNAC and a membrane, the authors used a molar ratio of 400:1 of SNAC:semaglutide to model approximately the ratio found in the prescription tablets of semaglutide. However, the aqueous molar concentration of SNAC in the periodic box probably far exceed the concentration relevant to its interaction inside the stomach. The ratio of SNAC molecules to phospholipids in the box is quite high as well and that can contribute to the ease of aggregation observed in the simulations in both the water and membrane phases. Maybe the authors can comment on those possible concentration differences of their simulation systems compared to the real-life environment, and if that apparent discrepancy could modify their conclusions.
- (4) How the plots of Fig. 2B were obtained? Was the plot symmetrized around the membrane center? Usually, computations with umbrella sampling using a single reaction coordinate produce an asymmetric plot for membrane permeation of a large molecule.
- (5) In the paragraph before line 178, the authors mentioned Supplementary Fig. 13 when describing the ionization of SNAC in the water phase. That figure doesn't seem to be related with its ionization in the water phase.
- (6) The Potential of Mean Force for the insertion of the semaglutide lipid tail inside the POPC membrane (Supplementary Fig. 16) probably has not converged yet. Pulling a large chain containing polar groups inside the membrane usually produce hysteresis caused by slow degrees of freedom that are not enhanced during the umbrella sampling (for example, distortions of phospholipids from their regular positions, formation of water wires, etc.). Still, the observed PMF drop when the terminal carboxylic group is pulled close to the membrane center is probably correct (at least qualitatively). The magnitude of the barrier after that drop is probably less accurate.
- (7) The figure caption for the Supplementary Fig. 3 states that there is an "increase" in the apparent diffusion coefficient. I

think there is a decrease of the diffusion coefficient (as mentioned correctly in the main text).
(8) The x-axis values are shifted in Supplementary Fig. 19 compared to Fig. 2 in the main text.

Reviewer #2

(Remarks to the Author)

In general, it is known that the permeability enhancers in oral absorption cause reversible disorganization-fluidization of the epithelial plasma membrane or physical interaction with the active substance (e.g., hydrophobization) to ensure passive diffusion. However, there is still debate on the permeability enhancers' mechanism of action. SNAC is a small fatty acid derivative and has been shown to provide a local pH increase to protect semaglutide from proteolytic degradation in the stomach and to facilitate the absorption of semaglutide from the gastric epithelium, primarily via the transcellular pathway. SNAC's effect on the intestinal epithelium is thought to be due to increased lipophilicity resulting from its non-covalent macromolecule complexation with the active substance (Drugs, 2021, 81, 1003-1030; Pharmaceutics, 2019, 11, 78).

In this study, the authors aimed to investigate the molecular mechanism of the passage of semaglutide from the intestinal barrier upon its oral administration with a permeability enhancer, salcaprosate sodium (SNAC). Semaglutide is a polar peptide containing 31 amino acids similar to GLP-1; it is indicated for treating patients with type II diabetes whose blood sugar is inadequately controlled. Semaglutide is available in tablet dosage forms formulated with SNAC in doses of 3 mg, 7 mg, and 14 mg.

The authors reported that SNAC fluidizes the membrane, which then assists semaglutide in passing through the gastric epithelial cells. Furthermore, they stated that although initial computational studies have provided valuable insights into the interactions between SNAC and model membranes, no detailed molecular mechanism for semaglutide permeation has yet been verified, and specifically, it remains unclear how membrane fluidization can facilitate peptide permeation without damaging the epithelial cells.

For this purpose, the authors used constant pH molecular dynamics (CpHMD) to predict the charges and interactions between semaglutide, SNAC, and the membrane by the GROMACS package software. They observed that SNAC formed aggregates around semaglutide inside and outside of the model membrane. The interactions between SNAC molecules in CDC13 as a well-established model for the hydrophobic membrane interior) were also determined by 1H NMR. Aggregate formation depending on SNAC concentration was verified by the increase in diameter determined by DLS. The presence of coaggregates of SNAC and semaglutide was demonstrated using 1H DOSY NMR which measures the diffusion coefficients. Based on computational and experimental results, SNAC-assisted permeation of semaglutide has been identified at the molecular level.

As a professional in pharmaceutical technology, I find this study results valuable. They confirm the effectiveness of permeability enhancers for the oral administration of active substances in peptide structure, a topic of great interest and relevance to our field.

However, experts' opinions in the related field are crucial to appraise the experimental procedures and simulations mainly carried out in this study.

Reviewer #3

(Remarks to the Author)

This study uses molecular dynamics simulations and NMR to speculate the possible mechanism by which the peptide drug semaglutide can permeate the membrane in the presence of the permeation enhancer SNAC. The topic is interesting and significant; however, the work appears premature and the conclusions are rather speculative. Several major concerns are discussed below.

- In part I (results shown in Figure 2), the authors calculated the free energy profile of a single SNAC passive permeation through the lipid bilayer. The initial frames for WHAM simulations were generated from pulling simulations and not shown in the main text. In fact, the free energy profile in Figure 2B does not even show SNAC. SNAC is very long; to this reviewer's knowledge there is no chance that the conformations of SNAC in the membrane can be realistically modeled using the standard approaches that the authors employed.
- In part I, panel B, the free energy profile is obtained at pH 5, which is not physiologically relevant.
- Figure 2 is confusing. For example, panel C) shows 50 NAC collapsed onto semaglutide, but B) actually shows the PMF of one NAC.
- In part II, the authors show two lines of experimental evidence: SNAC can aggregate (Fig. 3) and SNAC and semaglutide can co-aggregate (Table 1); however, experiment does not address whether they together permeate the membrane.
- In part III, the authors simulated the aggregation NAC and co-aggregation of NAC/semaglutide. Based on the structures, I have no doubt aggregation and co-aggregate occur; however, the snapshots of one NAC and one semaglutide in the model membrane (Figure 4C) are rather arbitrary and does not provide support to author's claim.
- In the last part, Figure 5 appears to show significantly more (400) NAC than lipid molecules such that the model membrane collapses, rather than providing support for the claim of pore formation.

Reviewer #4

(Remarks to the Author)

The present study concerns the mechanism of how peptide drugs together with permeation enhancers permeate passively through membranes. Specifically, the peptide semaglutide and the enhancer SNAC were investigated. The authors use mainly constant pH MD simulations to investigate the process to account for environment-dependent protonation of peptide and SNAC molecules. Additionally experimental methods are used (solution NMR, incl. DOSY; DLS) to corroborate the findings.

The authors come up with a plausible model to explain their computational and experimental findings, which is summarized in Fig. 6. This model which is reminiscent of a "quicksand-like" effect seems likely and is therefore an excellent basis for future studies that can put this model to the test. While the model appears valid it would not be unexpected if it needed to be refined and modified in the future.

The paper is very clearly written and easy to follow. However, sometimes it appears a bit superficial and I would recommend to include more details. For instance:

p.8 "significantly lowers the pKa value"- please indicate the value
or on p.10 "...the observed upfield shift" - please indicate the observed shift
etc...

The authors performed NMR and DLS experiments in CDC13 as a mimic of the hydrophobic membrane interior. Why can't the experiments be done in a more native-like environment, e.g. by solution NMR in detergent micelles or by solid-state NMR in liposomes? Would this be a perspective for future studies?

The idea to study the system of interest by a combined MD simulation / experimental approach is excellent but in this case would benefit from a slightly stronger focus on the experiments. Nevertheless, the developed model seems a very good basis to design future studies and in summary I therefore recommend publication of this paper in Nature Communications.

Version 1:

Reviewer comments:

Reviewer #1

(Remarks to the Author)

In this revised version, the authors addressed the more important concerns I had in the original version regarding the simulation part. As a result, their conclusions are more solidly grounded and convincing.

After reading this revised version, I have only minor observations and found a few typos:

1. In panel D of Figure 2, there are labels 1 and 3, that I think refer to the configurations depicted in panel C. If that is the case, the caption for panel D should be corrected to "SNAC outside (1) and inside (3).
2. In page 8, line 179: glyercol -> glycerol.
3. In Figure 5, panel B, the authors should provide a brief description of what the different molecular colors represent.
4. Looking at the configurations shown in Supplementary Fig 22 and associated movie 6 that are related to the computation of the PMF of the semaglutide lipid tail, it seems that the second end of the tail was restrained outside the membrane (that end doesn't move much in the movie). If such a restraint was applied, that should be mentioned in the text or Supplementary information.
5. In the Methods section, line 419, describing the larger box with four micelles, why the concentration of SNAC double to 80 mM and the concentration for CTAB remained the same compared to the concentrations for the smaller boxes? Any changes of concentration should be the same for both species.
6. In the Supplementary Information page S2, it says "32 POPC lipids per leaflet". I guess that number should be updated.

Reviewer #2

(Remarks to the Author)

My concerns have been addressed in the revision.

Reviewer #3

(Remarks to the Author)

The authors have addressed my concerns and comments.

Reviewer #4

(Remarks to the Author)

The authors have addressed my concerns adequately and I support now publication of the manuscript in its current state.

Point-by-point response to the referee reports regarding our manuscript entitled “Permeation Enhancer-Induced Membrane Defects Assist the Oral Absorption of Peptide Drugs”

We thank all four reviewers for their thoughtful and constructive comments. We have tried to address all of the comments/suggested changes and have highlighted all our changes in our manuscript and supporting information with a yellow background. Please find our detailed response to all the reviewer comments below:

Reviewer 1. We thank Reviewer 1 for their constructive review. Below, we have outlined in detail the changes we made to our revised manuscript in an effort to incorporate all the suggestions:

- 1.) Quoting Reviewer 1: “*The simulation protocols followed in the work are adequate. The cpHMD tool added to Gromacs is a very recent development that have been used mostly to study protonation states of peptides and proteins in aqueous environments. I wonder if the protocols described by the designers of the phbuilder package to parametrize the coefficients for pH control work correctly in a membrane environment given that the runs used for parametrization of the dVdl coefficients contains the charge groups inside a water box. Maybe the authors can comment on that possible issue.*”

We thank reviewer 1 for raising this important question. We have addressed it by modeling the assembly of oleic acid micelles in different pH environments to demonstrate that the CpHMD tool can adequately model dynamic protonation events in membrane-like environments. First, oleic acid was parametrized in water (with 0.15 M NaCl and 1 buffer particle), analogous to how we parametrized SNAC. Next, 10 molecules of oleic acid were randomly assembled in a 7.5 nm³ box and simulated for 100 ns ($n = 3$) at different pH values to create the theoretical titration curve shown below ($pH_a^{\text{theo}} = 5.8$), which matches reasonably well with the experimentally observed apparent pK_a of oleic acid/glycerol monooleate systems ($pK_a = 6.0$, reported in: Salentinig *et. al. Langmuir* **2016**, 26, 11670–11679). These new simulation results have been added as a new paragraph in the main text in the “Results” section:

Additional Paragraph Added to Main Text:

“To demonstrate that the CpHMD model applied in this work can reproduce the pK_a shifts of carboxylic acids in membrane-like environments, we applied the CpHMD model to a well-established oleic-acid model system. Self-assembled oleic acid demonstrates pH-dependent structures and composition-dependent apparent pK_a values,³⁵ with the apparent

pK_a value of oleic acid/glycerol monooleate systems ($pK_a \geq 6$) generally increasing with increasing oleic acid concentration. This experimentally-demonstrated³⁵ pK_a trend (with oleic acid in micelles being significantly less acidic than oleic acid in its monomeric form ($pK_a = 4.8$)) thus represents a well-established model system, which enabled us to validate the CpHMD model for carboxylic acids structurally related to SNAC in a membrane-like environment with experimental pK_a data. Overall, we found that performing a theoretical titration (Supplementary Fig. 31) of oleic acid — with $\partial V/\partial \lambda$ CpHMD coefficients obtained from simulations of monomeric oleic acid in water with 0.15 NaCl (see also Supplementary Figs. 14–15) — reproduces the experimentally observed shift in pK_a value reasonably well. Specifically, 10 molecules of oleic acid were randomly assembled in a 7.5 nm³ box and simulated for 100 ns ($n = 3$) at different pH values to create the theoretical titration curve shown in Supplementary Fig. 31 ($pK_a^{\text{theo}} = 5.83$, fit with a Hendersson-Hasselbach model), which matches with the experimentally observed apparent pK_a value of oleic acid/glycerol monooleate systems ($pK_a = 5.98$).³⁵ Overall, these results serve as proof of concept that the CpHMD methodology applied in this work can capture the influence of membrane-like environments on the pK_a of hydrophobic carboxylic acid embedded in a membrane.”

The corresponding supporting data has also been added to the Supporting Information file (see: Supplementary Figs. 14, 15, and 31, and Table S1, copied below for convenience as well):

Newly Added Supplementary Figures:

Supplementary Fig. 14 | Fit for the $\partial V/\partial\lambda$ CpHMD Coefficients for the OLET Residue.

Supplementary Fig. 15 | Distribution of λ -Values Plotted for 10 100 ns Replica Validation Simulations of the OLET Residue.

Supplementary Fig. 31 | Validation of the CpHMD Approach in a Membrane-like Environment by Predicting the Apparent pK_a Value of an Oleic Acid Micelle Model System. (A) Oleic acid (10 molecules, 40 mM) was randomly placed in a water box with 100 buffer particles and allowed to self-assemble for 100 ns ($n = 3$) in different pH environments ranging from pH 3 to 9 (solvent and buffer molecules are not shown in the figure for clarity). The $\partial V/\partial \lambda$ CpHMD coefficients for oleic acid were obtained from simulations of monomeric oleic acid in water (see also Supplementary Figs. 14 and 15) containing 0.15 M NaCl. (B) All replicas of this simulation resulted in the formation of micelles. The distribution of λ -coordinates was determined and averaged across all three replicas to produce a titration curve, which was then fit using the Henderson-Hasselbalch equation. The apparent pK_a of this simplified system ($pK_a^{\text{theo}} = 5.8$) matches reasonably well with the apparent pK_a of oleic acid/glycerol monooleate systems containing 22% w/w oleic acid ($pK_a = 6.0$). This serves as proof of concept for how this methodology can accurately model the influence of a membrane-like environment on the pK_a values of carboxylic acids.

Appended to Supplementary Table 1:

[OLET]							
incl = OLE OLEH							
atoms = H1 O1 O2 C1 C2							
qqA = 0.440 -0.610 -0.550 0.750 -0.210							
pKa_1 = 4.8							
qqb_1 = 0.000 -0.760 -0.760 0.620 -0.280							
dvd1_1 = -43.047 213.875 -336.373 -103.767 397.505 -192.068 -555.38 22.658							
Additional CHARMM36 forcefield parameters for carboxylic acids							
i	j	k	l	func	phi0	cp	mult
CG321	CG2O2	OG311	HPG1	9	180.00	-6.0000	2
CG321	CG2O2	OG311	HPG1	9	180.00	1.5000	4

- 2.) Quoting Reviewer 1: “Simulations to compute Potential of Mean Force for insertion of SNAC or the semaglutide tail with umbrella sampling used membrane patches of 32 POPC phospholipid molecules per leaflet. The lateral box size is probably small and that could produce some simulation artifacts. Why did the authors use that small membrane patches to study permeation of those two relatively long molecules?”

We thank the reviewer for pointing out the limited membrane size of our original simulations for the WHAM analyses. To address this concern, we have rerun our pulling simulations as well as all the WHAM analyses with larger membrane models containing 64 POPC phospholipids per leaflet. The new results, obtained with 64 POPC lipids per leaflet and with updated umbrella sampling windows, have been incorporated into the main text (Fig. 2B) and the Supplementary Information (Supplementary Figs. 18–24).

Revised Figures:

Fig. 2 | Impact of Variable Protonation on the Membrane Insertion of the Permeation Enhancer SNAC. (A) (...) (B) Potential of mean force (PMF) profiles (310.15 K, 0.15 M NaCl, $pH = 5.0$) for SNAC insertion into a POPC model membrane (64 lipids per leaflet). The PMF compares a standard, fixed-protonation-state model for protonated SNAC (blue curve) to the CpHMD model of SNAC (red curve). The free energy curve was obtained by umbrella sampling with harmonic restraints and a force constant of $1000 \text{ kJ mol}^{-1} \text{ nm}^{-2}$ as detailed in Supplementary Figs. 17 and 18, followed by standard WHAM analysis.¹ See Supplementary Movie 1 for the trajectory of the corresponding pulling simulation, which generated the initial frames for the WHAM windows. Shaded bands represent standard error estimates obtained with bootstrapping ($n = 200$) analysis implemented in the GROMACS WHAM analysis software. The distribution of lambda states at different z coordinates is shown with a grey dotted line to highlight the environment-dependent nature of SNAC protonation. These z -distance-dependent protonation probabilities were obtained from the CpHMD simulations by binning structures from select WHAM windows based on their z -positions and dividing the number of deprotonated frames by the total number of frames in each bin. Additionally, representative structures at key positions, both inside and outside of the membrane, are numbered and shown in panel (C).

Supplementary Fig. 20 | Snapshot of the SNAC in a POPC Membrane Model Described in Fig. 2B (z-Position = -1.1 nm). The corresponding trajectory for this process is shown in Supplementary Movie 1.

Supplementary Fig. 21 | Potential of Mean Force (PMF) Profile for Insertion of the Semaglutide Lipid Tail into a POPC Model Membrane. The PMF was obtained at 310.15K with the semaglutide lipid tail modeled as compound S1 with the CHARMM36 Force Field. The water layer contained 0.15 M NaCl and the membrane was built with 64 POPC lipids per leaflet. The free energy curve was obtained by umbrella sampling (with the z-distance from the POPC lipid head groups of the carboxylic acid end tail (highlighted in red) of the semaglutide tail constrained to the Umbrella sampling windows shown in Supplementary Fig. 22, followed by standard WHAM analysis. Harmonic restraints with a force constant of $1000 \text{ kJ mol}^{-1} \text{ nm}^{-2}$ were used for all umbrella sampling windows. Shaded bands represent standard error estimates obtained with bootstrapping analysis ($n = 200$) implemented in the GROMACS WHAM analysis software.¹²

Supplementary Fig. 22 | Representative Snapshots for the PMF Profile Shown in Supplementary Fig. 21. The z-distance from the membrane center (COM of the phospholipid headgroups) of the COOH anchor (the atom highlighted in red in Supplementary Fig. 21) is labeled for each snapshot. For a video of the corresponding trajectory, see Supplementary Movie 6.

Supplementary Fig. 23 | λ -Coordinate Trajectory for Pulling Semaglutide's Lipid Tail into the Membrane. See Supplementary Fig. 22 for corresponding simulation snapshots. While insertion of the lipid tail into the membrane is favorable overall as shown by the PMF in Supplementary Fig. 21, pulling semaglutide's lipid tail completely to the other side of the membrane positioned the ionizable site of the gGlu-2xOEG linker inside the lipid bilayer, which ultimately lead to an increase of the free energy by $\sim 8 \text{ kcal mol}^{-1}$ toward the end of the PMF profile shown in Supplementary Fig. 21.

Supplementary Fig. 24 | Umbrella Sampling Windows used to Calculate the PMF Profile Shown in Supplementary Fig. 21. This data was obtained with the CpHMD model as detailed in the Supplementary Methods section. Sampling windows were collected for 200 ns with harmonic restraints and a force constant of $1000 \text{ kJ mol}^{-1} \text{ nm}^{-2}$.

- 3.) Quoting Reviewer 1: *“In the simulations that contain semaglutide, SNAC and a membrane, the authors used a molar ratio of 400:1 of SNAC:semaglutide to model approximately the ratio found in the prescription tablets of semaglutide. However, the aqueous molar concentration of SNAC in the periodic box probably far exceeds the concentration relevant to its interaction inside the stomach. The ratio of SNAC molecules to phospholipids in the box is quite high as well and that can contribute to the ease of aggregation observed in the simulations in both the water and membrane phases. Maybe the authors can comment on those possible concentration differences of their simulation systems compared to the real-life environment, and if that apparent discrepancy could modify their conclusions.”*

We thank the reviewer for raising this important point. We believe that, while the concentration of SNAC in our system is relatively high, it can manifest the high local concentrations arising in the direct vicinity of a Rybelsus® (semaglutide) tablet, while at the same time reflecting the molar ratio of peptide/permeation enhancer of the tablet. Specifically: A Rybelsus® (semaglutide) tablet contains between 7–14 mg of semaglutide (1.7–3.4 μmol) — depending on the dose — with 400 mg of SNAC (1.3 mmol), translating into a $\sim 1:760$ to $\sim 1:380$ semaglutide:SNAC ratio. Our simulations used 400 SNACs per semaglutide molecule in the simulation box, which accurately reflects the experimental ratio of permeation enhancer to peptide drug. Furthermore, it is generally believed that

locally, i.e., directly around the tablet, the SNAC concentration is increased significantly compared to what would be expected after even distribution in the stomach, since the tablet is located directly on the stomach epithelia — as described, for example, by Dr. Brayden’s work: See, e.g.: Brayden *et al.*, *Pharmaceutics* **2019**, *11*, 78 (Ref. 42 in our manuscript). A prior study by Novo Nordisk (see: Knudsen *et al.*, *Sci. Transl. Med.* **2018**, *10*, 467, eaar7047/7041, Ref. 5 in our revised manuscript) also measured the SNAC concentration directly underneath the tablet and found the local SNAC concentration to be ~280 mM, which is reasonably well represented by our simulations, in which the overall SNAC concentration was ~350 mM. Furthermore, the semaglutide concentration in our simulations (~800 μ M) is also reasonable and close to what has been reported experimentally (~600 μ M) by Knudsen *et al.* (See: *Sci. Transl. Med.* **2018**, *10*, 467 eaar7047/7041) in the direct vicinity of the tablet.

The following sentence is present in the main text, to explain our choice of the amount of SNAC added: “The 400:1 molar ratio of SNAC:semaglutide in the simulations was chosen based on the current formulations in Rybelsus,⁴¹ which contains ~1 mmol of SNAC per tablet in a ~380:1 to ~760:1 SNAC:semaglutide molar ratio.”

Furthermore, we have added the following paragraph to our revision, to further clarify our choice of the SNAC concentration in the simulation box:

Additional Text Added to Main Text:

“The composition of the system was based on the formulation of the Rybelsus tablet⁴¹ and the fact that the localized concentration of SNAC around the tablet is significantly higher than what would be expected after even distribution in the stomach, as the tablet is located directly on the stomach epithelia.⁴² The concentrations of SNAC and semaglutide at the site where the tablet is located in the stomach have been measured at roughly 280 mM and 600 μ M, respectively.⁵ To match these relatively high concentrations localized under the tablet, semaglutide (800 μ M) and SNAC (350 mM) were randomly placed around the membrane and the system was run for 100 ns to equilibrate.”

- 4.) Quoting Reviewer 1: “*How the plots of Fig. 2B were obtained? Was the plot symmetrized around the membrane center? Usually, computations with umbrella sampling using a single reaction coordinate produce an asymmetric plot for membrane permeation of a large molecule.*”

In our original submission, the plot was symmetrized around the membrane center, as SNAC was initially only pulled from the membrane center into the water phase. In our revision, to confirm the symmetric nature of this PMF, we have now repeated the umbrella sampling with regular MD, pulling SNAC fully through the membrane—from the water

layer into the center of the membrane and then out into the water layer on the other side. See Supplementary Movie 1 for the trajectory of the new pulling simulation, which provided the starting structures for the umbrella sampling windows. To further improve the accuracy of this simulation, we also increased the system size to 64 lipids per leaflet (as discussed above), increased the number of umbrella sampling windows to 72, and extended the sampling length to 200 ns per window. In total, the combined simulation time for this system (covering all the umbrella sampling windows) now exceeds 14 μ s. New plots for the updated simulations are included in the main text (Fig. 2) and supporting information (Supplementary Figs. 16 and 17). The symmetry of the PMF (Supplementary Fig. 16) suggests that only the first half of the plot needs to be assessed for *CpHMD* umbrella sampling. Additionally, Fig. 2 has been expanded to include relevant structures of SNAC (snapshots from the MD simulations), both inside and outside the membrane, as well as normalized histogram plots of the distance between the aromatic ring and the carboxylic acid of SNAC. These plots compare *CpHMD* and standard MD methods and demonstrate that, with our methodology, all SNAC structures sample both extended and contracted conformations, addressing a significant concern of Reviewer 3 (*vide infra*).

New Supplementary Figures of the PMF (Not Symmetrized) with a single SNAC Molecule and the Corresponding Umbrella Sampling Windows:

Supplementary Fig. 16 | Potential of Mean Force (PMF) Profile for pulling a SNAC molecule through a POPC Model Membrane. The PMF was obtained at 310.15K with SNAC, and the membrane (containing 64 POPC lipids per leaflet) was modeled with the CHARMM36 Force Field. The water layer contained 0.15 M NaCl. The free energy curve

was obtained by umbrella sampling (with harmonic restraints on the z-coordinate — defined as the z-distance from the membrane center (COM of the phospholipid headgroups) to the COM of SNAC — with a force constant of $1000 \text{ kJ mol}^{-1} \text{ nm}^{-2}$) with the center of mass of SNAC constrained to the umbrella sampling windows shown in Supplementary Fig. 17, followed by standard WHAM analysis. Shaded bands represent standard error estimates obtained with bootstrapping analysis ($n = 200$) implemented in the GROMACS WHAM analysis software.¹²

Supplementary Fig. 17 | Umbrella Sampling Windows used to Calculate the PMF Profile Shown in Supplementary Fig. 16. This data was obtained with standard molecular dynamics with window sampling lengths of 200 ns with harmonic restraints and a force constant of $1000 \text{ kJ mol}^{-1} \text{ nm}^{-2}$. Initial conformations were obtained from the pulling simulations shown in Supplementary Movie 1.

- 5) Quoting Reviewer 1: *"In the paragraph before line 178, the authors mentioned Supplementary Fig. 13 when describing the ionization of SNAC in the water phase. That figure doesn't seem to be related with its ionization in the water phase."* We thank the reviewer for catching this error. The incorrect supplementary figure was referenced in our original submission. This error has been corrected in our revised manuscript to now the correct figure (Supplementary Fig. 19).

- 6) Quoting Reviewer 1: *"The Potential of Mean Force for the insertion of the semaglutide lipid tail inside the POPC membrane (Supplementary Fig. 16) probably has not converged yet. Pulling a large chain containing polar groups inside the membrane usually produce hysteresis caused by slow degrees of freedom that are not enhanced during the umbrella sampling (for example, distortions of phospholipids from their regular positions, formation of water wires, etc.). Still, the observed PMF drop when the terminal carboxylic group is pulled close to the membrane center is probably correct (at least qualitatively). The magnitude of the barrier after that drop is probably less accurate."*

This is a good point. To clarify the convergence of this simulation, it has now been rerun with longer umbrella sampling windows (200 ns per window) and a larger membrane model (64 POPC lipids per leaflet), as shown in Supplementary Figs. 21–24. As the reviewer suspected, the results with this bigger system are qualitatively very similar to those previously obtained.

- 7) Quoting Reviewer 1: *"The figure caption for the Supplementary Fig. 3 states that there is an "increase" in the apparent diffusion coefficient. I think there is a decrease of the diffusion coefficient (as mentioned correctly in the main text)."*

We sincerely thank the reviewer for pointing out this mistake. It has been corrected. The updated figure caption is provided below for reference:

"Supplementary Fig. 3 | Stacked Convection-compensated ¹H DOSY NMR Spectra (800 MHz, CDCl₃, 298 K) of SNAC at 5 mM and 100 mM Concentrations. A clear **decrease** in the apparent diffusion coefficient is observed with increasing SNAC concentration, consistent with the formation of dimers and larger oligomers observed by DLS (Supplementary Fig. 4) and all-atom CpHMD simulations (Supplementary Fig. 26)."

- 8) Quoting Reviewer 1: *"The axis values are shifted in the Supplementary Fig. 19 compared to Fig. 2 in the main text."*

We thank the reviewer for bringing this discrepancy to our attention. The referenced figures have all been adjusted so that the x-axes now all match for all corresponding figures. Specifically, Figs. 2B and Supplementary Figs. 16, 17, 18, 19, 21, and 24 have all been revised accordingly.

Reviewer 2. We thank Reviewer 2 for their constructive review.

- 1) Quoting Reviewer 2: *"As a professional in pharmaceutical technology, I find this study's results valuable. They confirm the effectiveness of permeability enhancers for the oral administration of active substances in peptide structure, a topic of great interest and relevance to our field."*

We appreciate the reviewer's feedback, which highlights the importance and general interest of this work.

Reviewer 3. We thank Reviewer 3 for providing detailed responses and offering valuable feedback on our manuscript. Below, we have outlined in detail the changes we made to our revised manuscript in an effort to incorporate most of the suggestions:

- 1) Quoting Reviewer 3: *“In part I (results shown in Figure 2), the authors calculated the free energy profile of a single SNAC passive permeation through the lipid bilayer. The initial frames for WHAM simulations were generated from pulling simulations and not shown in the main text. In fact, the free energy profile in Figure 2B does not even show SNAC.”*

We thank the reviewer for their suggestion and have now added snapshots of SNAC in the model membrane system. This is clearly shown and labeled in Fig. 2C of the main text in our revised manuscript. Additionally, we have now also included two example trajectories from the umbrella sampling simulations (see Supplementary Movies 16 and 17). These display not only the initial frames (obtained from pulling simulations) but also the effective sampling in both the water layer (Supplementary Movie 16) and in the membrane (Supplementary Movie 17).

Revised Figure:

Fig. 2 | Impact of Variable Protonation on the Membrane Insertion of the Permeation Enhancer SNAC. (A) Definition of the λ -coordinate in SNAC for the C_pHMD model. (B) Potential of mean force (PMF) profiles (310.15 K, 0.15 M NaCl, pH = 5.0) for SNAC insertion into a POPC model membrane (64 lipids per leaflet). The PMF compares a standard, fixed-protonation-state model for protonated SNAC (blue curve) to the C_pHMD model of SNAC (red curve). The free energy curve was obtained by umbrella sampling with harmonic restraints and a force constant of 1000 kJ mol⁻¹ nm⁻² as detailed in Supplementary Figs. 17 and 18, followed by standard WHAM analysis.¹ See Supplementary Movie 1 for the trajectory of the corresponding pulling simulation, which generated the initial frames for the WHAM windows. Shaded bands represent standard error estimates obtained with bootstrapping ($n = 200$) analysis implemented in the GROMACS WHAM analysis software. The distribution of lambda states at different z coordinates is shown with a grey dotted line to highlight the environment-dependent nature of SNAC protonation. These z-distance-dependent protonation probabilities were obtained from the C_pHMD simulations by binning structures from select WHAM windows based on their z-positions and dividing the number of deprotonated frames by the total number of frames in each bin. Additionally, representative structures at key positions, both inside and outside of the membrane, are numbered and shown in panel (C). To evaluate how effectively different conformations of SNAC are being sampled using C_pHMD as compared to standard methods, the distances between the aromatic ring and carboxylic acid of SNAC were plotted as normalized histograms. (D) Superimposed histogram plots with C_pHMD on (blue) and off (red) for SNAC inside (1) and outside (3) of the membrane show that both methods readily sample extended and contracted conformations of SNAC inside and outside the membrane. Representative coiled and linear structures of SNAC are included with arrows pointing to their distance on each graph. SNAC favors a more coiled structure inside the membrane as compared to outside the membrane, which is reflected with both methods. The trajectories of these two example umbrella sampling windows are shown in Supplementary Movies 16 and 17.

- 2) *Quoting Reviewer 3: "SNAC is very long; to this reviewer's knowledge there is no chance that the conformations of SNAC in the membrane can be realistically modeled using the standard approaches that the authors employed."*

We appreciate the reviewer's concern about the conformational sampling of SNAC in the membrane. Previously published work (See: Sun *et al.*, *J. Phys. Chem. B* **2024**, *128*, 1668, Ref. 23 in our manuscript) has provided the free energy profile of SNAC insertion with an alternative 2D-metadynamics-based enhanced sampling technique for protonated SNAC. We compared these published results to our methodology and were pleased to find that our standard umbrella sampling results agree reasonably well with the previously published work as shown below:

Our results being in qualitative agreement with what has previously been reported, taken together with the fact that we obtained a symmetric free-energy profile for SNAC entering/exiting the membrane (see: Supplementary Fig. 16), with each half of the free energy profile obtained from different pulling conformations and independent umbrella-sampling windows (see also Supplementary Movie 1: The left-side of the PMF was generated by SNAC being pulled from the outside of the membrane to the inside, while the right side of the PMF was generated with the opposite process, i.e., by pulling SNAC from the center of the membrane to the water phase) indicates that our classical umbrella sampling approach is indeed able to efficiently sample the different conformations of SNAC inside the membrane. While some quantitative differences remain between the two models, we do not believe that these differences are due to insufficient sampling. To illustrate this point, we further analyzed the conformations sampled by SNAC inside and outside of the membrane (Fig. 2D, panel copied below as well for convenience) and found that SNAC readily samples both extended and contracted conformations inside the membrane within the timescale of our 200 ns WHAM windows, clearly refuting the reviewer's claim that there is "no chance that long SNAC can be properly sampled inside the membrane" with our methodology. Rather, we would like to point out that there are other significant differences between the two models (beyond the different enhanced sampling methodologies used), which could well be responsible for the relatively small remaining quantitative differences in the free energy plots obtained. Notably, the membrane model employed in Ref. 23 used only 20 POPC lipids per leaflet, which — as

indicated by reviewer 1 — could “lead to simulation artifacts due to small system size”. In turn, in our revised model, we have now used 64 POPC lipids per leaflet in an effort to minimize such finite-system artifacts.

New Figure Panel Added to Fig. 2 Demonstrating Effective Sampling of SNAC Both Inside and Outside of the Membrane:

- 3) Quoting Reviewer 3: “In part I, panel B, the free energy profile is obtained at pH 5, which is not physiologically relevant.”

We disagree with this comment, as outlined in detail below. However, first of all, the reviewer does not specify what they would consider to be an “appropriate” physiological pH for us to perform our simulations at. Thus, we interpret this comment based on previously published work (for example, Ref. 23), where permeation enhancers were treated in their protonated form. We are assuming that, based on this previously published work, the reviewer might feel that we should simulate everything at a much more acidic pH to simulate the acidic environment of the stomach.

However, what clearly speaks against this argument is (quoting from one of Dr. Brayden’s recent reviews — see: Brayden *et al.*, *Pharmaceutics* **2019**, 11, 78, Ref. 42 in our revised manuscript): “The optimum once daily tablet consists of 14 mg of semaglutide co-formulated with 300 mc of SNAC. After digestion, the tablet erodes rapidly in the stomach, resulting in the release of a **highly concentrated amount of SNAC that neutralizes the pH of gastric fluid in the immediate vicinity of the tablet to inactivate pepsin.**” Furthermore, as clearly confirmed in Ref. 5 cited in our manuscript, SNAC’s local effect on increasing

the pH around the SNAC/semaglutide tablet inhibits the conversion of pepsinogen to pepsin, thereby protecting the peptide in the immediate vicinity of the tablet from proteolytic degradation. As also shown in Ref. 5 (*Sci. Transl. Med.* **2018**, *10*, eaar7047/7041) and Ref. 20 (Sleisenger and Fordtran's *Gastrointestinal and Liver Disease*, 9th Ed., Volume 1, 2010, Pages 817–832.e7, Chapter 49: Gastric Secretion), this effect of stabilizing semaglutide against degradation by pepsin is generally believed to become effective at pH values ≥ 5 . Below pH 5, pepsin is quite effective at degrading semaglutide. Given all this published evidence for localized buffering action of SNAC in the direct vicinity of the tablet, we do not feel that this comment by Reviewer 3 is scientifically justified. Of course, the actual pH around the tablet will be highly dependent on the exact amount of stomach acid present in each individual patient (so there is expected high variability and there is likely not a single correct answer). Further factors that will influence the physiologically relevant pH around the tablet include, but are not limited to, the dissolution rate of the tablet, gastric motion, and potential leftover food present in the patient's stomach after incomplete fasting. Overall, the expected variability in local pH around a Rybelsus tablet makes it impossible to assign a precise pH value for the simulations to be run at, in our opinion. Nevertheless, based on the available literature, a pH of 5, which models the less acidic environment in the close vicinity of the SNAC/semaglutide tablet in the stomach, is a reasonable ballpark estimate, in our opinion, based on the available literature evidence presented.

To address this important comment directly in the paper as well, the following paragraph has been added to the main text:

Additional Text Added to Main Text:

“It is currently known that SNAC stabilizes semaglutide in the stomach by inactivating pepsin (which is achieved by increasing the local pH around the tablet¹⁹). This SNAC-induced local buffering action and associated stabilization from proteolytic degradation in the stomach is generally believed to become effective at or above pH 5,²⁰ which therefore represents a reasonable pH estimate for the local environment in the close vicinity of a Rybelsus tablet (with local SNAC concentrations around a Rybelsus tablet as high as ~280 mM).⁵”

- 4) Quoting Reviewer 3: “Figure 2 is confusing. For example, panel C) shows 50 SNAC collapsed onto semaglutide, but B) actually shows the PMF of one SNAC.”

We very much appreciate this suggestion and have now split this figure apart into two separate figures to enhance the clarity for the readers in our revised manuscript. The resulting new figure (Fig. 3) is also shown below for convenience, while the revised Fig. 2 is shown above.

- 5) Quoting Reviewer 3: “In part II, the authors show two lines of experimental evidence: SNAC can aggregate (Fig. 3) and SNAC and semaglutide can co-aggregate (Table 1); however, experiment does not address whether they together permeate the membrane.”

First, it is already well known from the literature (see, e.g., Ref. 5 in our manuscript) that semaglutide cannot permeate the membrane without SNAC (or another, alternative permeation enhancer). Furthermore, prior work has already shown that SNAC gets absorbed concurrently with semaglutide. See, e.g.: Knudsen *et al.*, *Sci. Transl. Med.* **2018**, *10*, eaar7047/7041 (Ref. 5 in our revised manuscript), with relevant Figs. 3B/C, copied below, which show the time-plasma-concentration curves for both semaglutide and SNAC after dosing semaglutide together with SNAC orally in healthy individuals ($n = 26$). These experimental results, which are already published in the literature, strongly suggest that SNAC and semaglutide are somehow interacting with each other in the membrane (as shown by our simulations), especially given the fact that prior work (Ref. 5) has also established that the primary pathway of absorption for the semaglutide/SNAC combination is very likely transcellular.

Second, Reviewer 3 states that we should show experimentally that “*SNAC and semaglutide together permeate the membrane*”, which, in other words, would mean that the reviewer would like us to show that after SNAC and semaglutide aggregate in the water layer, these aggregates enter/permeate the membrane together. However, as clearly outlined in our manuscript, we don't actually believe that the entire SNAC/semaglutide cluster needs to enter the membrane together (as seems to be indicated by the reviewer). The free energy barrier for this process would be too high. Rather, we believe that aggregation is dynamic, which means individual SNAC molecules can leave the aggregate in water one by one, and then enter the membrane, and reaggregate in the membrane to form new dynamic aggregates. Our DOSY NMR results also demonstrate the dynamic nature of the aggregates, which, on the NMR timescale, leads to different (time-averaged) diffusion bands for SNAC and semaglutide. In our revised manuscript, we also present additional experimental and computational data with micelle models, which further indicate that SNAC can aggregate in membrane-like environments, thereby supporting our mechanistic model, as shown in Figs. 4B–D and Fig. 5B. Overall, we believe that our experimental model systems, coupled with the experimental evidence already in the literature (which clearly shows that semaglutide can only permeate the membrane in the presence of SNAC), are in strong support of our mechanistic hypothesis.

- 6) Quoting Reviewer 3: “*In part III, the authors simulated the aggregation SNAC and co-aggregation of SNAC/semaglutide. Based on the structures, I have no doubt aggregation and co-aggregate occur; however, the snapshots of one SNAC and one semaglutide in the model membrane (Figure 4C) are rather arbitrary and does not provide support to author’s claim.*”

To address this comment, we have performed a more extensive analysis of the corresponding 1- μ s simulation (and its three additional replicas) to more clearly show the mobility of SNAC throughout these simulations in our revised manuscript. The z-diffusion of SNAC was calculated for each replica in 100 ns increments using the *msd* functionality in GROMACS 2022 together with the corresponding mean lipid/water-exposed surface area for each SNAC molecule (which is a measure for how deep a SNAC is buried inside a cluster) for each SNAC over this time period. The aggregated data across all four replicas is plotted in Fig. 5C. This data shows that there clearly are mobile SNAC molecules in the membrane SNAC clusters, and these mobile SNACs move significantly (relative to their initial positions in the SNAC clusters) during these μ s-long simulations. Moreover, the distribution of mobile SNACs in comparison to their surface area exposed to water/lipids

indicates that SNAC molecules can be mobile in any part of the cluster or the membrane. Therefore, this result supports our overall hypothesis that mobile SNACs can facilitate peptide permeation. Additionally, snapshots are now also shown in Fig. 5D, which clearly highlight how all SNAC molecules that are initially found within 7.5 nm of semaglutide move during the simulation.

Newly Added Figure:

7) Quoting Reviewer 3: “In the last part, Figure 5 appears to show significantly more (400) SNAC than lipid molecules such that the model membrane collapses, rather than providing support for the claim of pore formation.”

While — as discussed above — the SNAC concentration in the membrane is relatively high in our models to account for the fact that the local SNAC concentration in the direct vicinity of the tablet is generally believed to be ~280 mM, we have not seen any evidence of the membrane collapsing. The membrane is stable, and no significant amounts of water (besides a few water molecules that are part of the SNAC clusters inside the membrane) are present in the membrane, which would be indicative of potential membrane collapse. This is shown in the figure below (final snapshots of the 1 μ s simulations described in our manuscript, with all the water molecules shown in space-filling mode). The membrane is also stable in all the other replicas, none of which shows any evidence of membrane

collapse. Additionally, the included supplementary movies and other figures in the manuscript/SI further demonstrate the integrity of the overall membrane structure over the length of the simulations.

Newly Added Figure:

Supplementary Fig. 32 | Membrane Stability After 1 μ s of Unbiased Simulation. Snapshots of the last frames for unbiased 1 μ s replicas as described in Fig. 6 of the main text. Water molecules within 2.5 nm of the membrane are all shown in space-filling mode to highlight the stability of the membrane after 1 μ s of simulation time (peptide omitted for clarity). The relatively few water molecules in the membrane are incorporated into SNAC clusters (cyan), thereby maintaining the membrane's integrity throughout the simulation.

Reviewer 4. We thank Reviewer 4 for their insight and suggestions for improving this manuscript.

- 1) Quoting Reviewer 4: *"The authors come up with a plausible model to explain their computational and experimental findings, which is summarized in Fig. 6. This model which is reminiscent of a "quicksand-like" effect seems likely and is therefore an excellent basis*

for future studies that can put this model to the test. While the model appears valid it would not be unexpected if it needed to be refined and modified in the future.”

We appreciate this thoughtful comment. The following statement has been added to the conclusions of our revised manuscript to highlight this point.

Revised Text:

“Overall, this work serves as the basis for future models that will likely further refine this complex process, as our results provide a new, viable mechanism for cell permeation, helping to advance the field of oral peptide drug delivery, including potentially for macrocyclic peptides in the future.^{3, 67-69”}

- 2) Quoting Reviewer 4: *“The paper is very clearly written and easy to follow. However, sometimes it appears a bit superficial and I would recommend to include more details. For instance: p.8 “significantly lowers the pK_a value” — please indicate the value or on p.10 “... the observed upfield shift” — please indicate the observed shift etc...”*

We thank the reviewer for these suggestions. More in depth analysis has been performed to determine how interactions with cationic residues influence the local protonation state of SNAC molecules. The highlighted SNAC molecule (inset in Fig. 3 of the revised manuscript) forms a salt bridge that reduces the local pK_a of that SNAC molecule to ~3.4. This pK_a was determined from the distribution of λ values from a 10 ns sampling period in which that SNAC molecule was participating in the salt bridge by applying the Henderson-Hasselbach equation, utilizing the ratio of frames in which the highlighted SNAC is protonated or deprotonated over the sampling window. The following more precise language has been included in the main text to better describe this effect:

“For instance, a salt bridge was formed between a SNAC molecule and R36 in semaglutide, which stabilizes the deprotonated state of the corresponding SNAC, resulting in a significantly lower local pK_a of 3.4 (Fig. 3) of the corresponding SNAC molecule.”

The statement discussing the upfield shift has also been clarified in the main text with the following revised sentence:

“Specifically, the ~0.1 ppm upfield shift of the amide resonance with increasing SNAC concentration is attributed to an increase in hydrogen bonding, which deshields the amide and carboxylic acid peaks.^{38”}

Additionally, more detail was added to more clearly state the influence of CpHMD and how it stabilizes SNAC in the aqueous phase in Fig. 2 with the following revised sentence:

“This process lowers the free energy for SNAC in the aqueous phase by ~1 kcal/mol, compared to (...)”

- 3) Quoting Reviewer 4: “*The authors performed NMR and DLS experiments in CDCl₃ as a mimic of the hydrophobic membrane interior. Why can't the experiments be done in a more native-like environment, e.g., by solution NMR in detergent micelles or by solid-state NMR in liposomes? Would this be a perspective for future studies?*”

We appreciate the reviewer's suggestion to study more relevant model systems to support this work. Solid-state NMR is a promising perspective for future studies; however, for our revised manuscript, we have now also performed NMR and DLS experiments with a detergent micelle model system (based on CTAB micelles) as membrane mimics, as recommended by the reviewer, to probe the interactions between a detergent (CTAB) and SNAC. The main text (Figs. 4 and 5) and the supplementary information (Supplementary Figs. 28–30, and Supplementary Movies 7–9) have been added/amended as described below to include the new results. These additional NMR, DLS, and theoretical simulations broadly support the aggregation of SNAC in/around CTAB micelles, which further supports our proposed SNAC absorption mechanism. The following text has been added to the manuscript to describe these results.

Newly Added Paragraph:

“Finally, the aggregation of SNAC in a membrane-like environment was also demonstrated using a detergent micelle model (CTAB), as evidenced by both DLS and NMR experiments (Figs. 4B–D). From these experiments, we found that the size of the SNAC/CTAB aggregates observed with DLS is sensitive to the SNAC concentration, resulting in aggregates of increasing size and complexity. At the same time, ¹H NMR titrations of SNAC into solutions of CTAB (Supplementary Fig. 28) demonstrate significant broadening of SNAC and CTAB resonances, which also supports their aggregation into larger structures. Furthermore, ¹H-¹H NOESY NMR experiments exhibited concentration-dependent SNAC cross-peaks. Specifically, we found the aromatic SNAC/SNAC NOE coupling intensities to be highly sensitive to the increasing SNAC concentration, suggesting that SNAC aggregates in the CTAB micelle environment, likely via a combination of π -stacking between the aromatic SNAC residues and hydrogen bonding interactions. This finding is also supported by CpHMD simulations of SNAC with CTAB micelles (Fig. 5B and Supplementary Fig. 30), which clearly show π -stacking between the corresponding ¹H NMR signals, for which the NOE cross peaks initially increase rapidly and non-linearly with increasing SNAC concentration, consistent with SNAC aggregation in/around the micelles. In turn, if SNAC were getting evenly distributed in/around the CTAB micelles (i.e., not aggregating in/on the micelles), the NOE cross-peaks between the

aromatic SNAC residues would be expected to increase at a similar rate as the CTAB/SNAC NOE cross-peaks. In contrast, the relative CTAB/SNAC NOE coupling intensity of the very clearly defined ω -H^{AC} crosspeak was found to increase only very slowly and in a primarily linear fashion with increasing SNAC concentration (Fig. 4D). We believe that this relatively small increase in CTAB/SNAC NOE cross-peak intensity is caused by the naturally more intense SNAC ¹H NMR signals arising with increasing SNAC concentration. Moreover, the relatively small change in NOE intensity observed for the CTAB/SNAC ω -H^{AC} NOE crosspeak also indicates that aggregation size effects on the correlation times are very likely small compared to the changes in NOEs caused by SNAC aggregation in our system, consistent with the fact that the large CTAB/SNAC aggregates (Fig. 4B) all lie in the slow-tumbling regime for NOESY NMR (with all NOE cross peaks for SNAC and CTAB being negative, i.e., with the same sign as the diagonal peaks as clearly shown in Fig. 4C and Supplementary Fig. 28B). Overall, the experimentally observed aggregation of SNAC in non-polar environments supports our computational results, which show that dynamic SNAC aggregates can form in CH₂Cl₂, in detergent micelles, and in the interior of a lipid bilayer membrane. At the same time, the dynamic nature of the SNAC aggregates in the membrane is demonstrated in Figs. 5C/D, Supplementary Figs. 35, and in Supplementary Movies 12–14, which all highlight the significant movement of selected SNAC molecules in the membrane (all of them part of SNAC membrane clusters) during the 1- μ s-long CpHMD membrane simulations.”

Newly Added Figures:

Supplementary Fig. 28 | Concentration-dependent NMR Data for CTAB Micelles and SNAC. (A) Stacked ^1H NMR spectra (500 MHz, D_2O , 298 K, referenced to the DSS internal standard) for CTAB (50 mM) and SNAC in different concentrations, with all NMR resonances assigned. The broadening and shifting of the SNAC resonances in the aromatic region, as well as of the CTAB resonances in the aliphatic region, support the computationally observed (see Fig. 5) aggregation of CTAB with SNAC, which is also reminiscent of what was previously observed for CTAB interacting with *p*-toluenesulfonate/*p*-toluenesulfonic acid.¹³ (B) Full ^1H - ^1H NOESY NMR spectrum (500 MHz, D_2O , 298 K) of 40 mM SNAC and its conjugate acid (4:1 molar ratio) and 50 mM CTAB.

Supplementary Fig. 29 | Images to Highlight the Increased Solubility of the Conjugate Acid of SNAC in the Presence of CTAB Micelles. (A) SNAC and its conjugate acid (4:1 molar ratio, 40 mmol total SNAC) were dissolved in 2.0 mL of MeOH. (B) A thin layer of SNAC film is formed in the scintillation vial after removing MeOH under reduced pressure. (C) Adding 1.0 mL of DI water to the SNAC film results in a cloudy solution due to the low solubility of the conjugate acid of SNAC in water. (D) Adding 1.0 mL of a 50 mM solution of CTAB in DI water results in a clear solution. Therefore, the presence of CTAB micelles increases the solubility of the conjugate acid of SNAC in an aqueous environment, consistent with our computational models (Fig. 5 and Supplementary Fig. 30), which also show SNAC and CTAB micelles coaggregating spontaneously in an aqueous environment.

Supplementary Fig. 30 | Snapshots of MD simulations of CTAB and SNAC Demonstrate the Aggregation of SNAC in CTAB Micelles. (A) Standard MD simulation of 25 molecules of CTAB (100 mM) forming a micelle. (B) *CpHMD* ($pH = 5.6$) simulation of 20 molecules of SNAC (blue: 40 mM) randomly placed around the CTAB (red: 25 molecules, 50 mM) micelle demonstrates the incorporation of SNAC into the CTAB micelle. (C) *CpHMD* ($pH = 5.6$) of four CTAB/SNAC micelles (circled in black) shows two micelles coming together to form a larger micelle (circled in red) that demonstrates SNAC clustering in the micelle. For a zoomed-in image of the final, assembled micelle, see Fig. 5B. These processes are also visualized in Supplementary Movies 7–9.

Point-by-point response to Reviewer 1 regarding our manuscript entitled “Permeation Enhancer-Induced Membrane Defects Assist the Oral Absorption of Peptide Drugs”

We have incorporated the minor changes that have been suggested by Reviewer 1 as detailed below:

- 1.) Quoting Reviewer 1: “*In panel D of Figure 2, there are labels 1 and 3, that I think refer to the configurations depicted in panel C. If that is the case, the caption for panel D should be corrected to “SNAC outside (1) and inside (3).”*”

We thank the reviewer for bringing this mistake to our attention. The figure caption has been updated as suggested by Reviewer 1.

- 2.) Quoting Reviewer 1: “*In page 8, line 179: glyercol -> glycerol.*”

This typo has been corrected in the newest version of the manuscript.

- 3.) Quoting Reviewer 1: “*In Figure 5, panel B, the authors should provide a brief description of what the different molecular colors represent.*”

A clear description of the colors used in Figure 5, panel B, has been added to the figure caption for clarity.

Added caption text:

“SNAC molecules are shown with carbons in opaque cyan, polar hydrogens in white, nitrogens in dark blue, and oxygens in red. CTAB is shown in blue in semi-transparent mode. Key hydrogen atoms observed in the ^1H - ^1H NOESY NMR experiments are highlighted in space-filling mode with H-*d* shown in magenta, and H-*b* in light green.”

- 4.) Quoting Reviewer 1: “*Looking at the configurations shown in Supplementary Fig 22 and associated movie 6 that are related to the computation of the PMF of the semaglutide lipid tail, it seems that the second end of the tail was restrained outside the membrane (that end doesn't move much in the movie). If such a restraint was applied, that should be mentioned in the text or Supplementary information.*”

We appreciate this suggestion. The reviewer is correct in that restraints were applied during the pulling simulation to generate the initial frames for the umbrella sampling (but not during the umbrella sampling itself). This has been clarified in the caption of Supplementary Fig. 21 with the following text:

Added caption text:

“The methyl caps of the model tail of semaglutide were restrained with a $1000 \text{ kJ mol}^{-1} \text{ nm}^{-1}$ harmonic position restraint during the initial pulling simulation, which was removed during umbrella sampling. Harmonic restraints with a force constant of $1000 \text{ kJ mol}^{-1} \text{ nm}^{-2}$ were used for all umbrella sampling windows.”

- 5.) Quoting Reviewer 1: “*In the Methods section, line 419, describing the larger box with four micelles, why the concentration of SNAC double to 80 mM and the concentration for CTAB remained the same compared to the concentrations for the smaller boxes? Any changes of concentration should be the same for both species.*”

The reviewer is correct, and we are grateful to the reviewer for pointing out this mistake in the manuscript (which had arisen as part of a copy-paste error). The concentration of SNAC in the initial SNAC/CTAB micelle simulation was corrected from “100 mM” to “50 mM” in the Methods section, which is the actual concentration we simulated.

- 6.) Quoting Reviewer 1: “*In the Supplementary Information page S2, it says ‘32 POPC lipids per leaflet’. I guess that number should be updated.*”

This typo has been corrected to the correct value (64 POPC) as described in other parts of the text.